# Spatially resolved evaluation of Earth system models with satellite column averaged CO<sub>2</sub>

Bettina K. Gier<sup>1,2</sup>, Michael Buchwitz<sup>1</sup>, Maximilian Reuter<sup>1</sup>, Peter M. Cox<sup>3</sup>, and Pierre Friedlingstein<sup>3,4</sup>, Veronika Eyring<sup>2,1</sup>

<sup>5</sup> <sup>1</sup>University of Bremen, Institute of Environmentral Physics (IUP), Bremen, Germany <sup>2</sup>Deutsches Zentrum für Luft- und Raumfahrt (DLR), Institut für Physik der Atmosphäre, Oberpfaffenhofen, Germany <sup>3</sup>College of Engineering, Mathematics and Physical Sciences, University of Exeter, Exeter, EX4 4QE, United Kingdom <sup>4</sup>LMD/IPSL, ENS, PSL Université, École Polytechnique, Institut Polytechnique de Paris, Sorbonne Université, CNRS, Paris, France

10 Correspondence to: Bettina K. Gier (gier@uni-bremen.de)

Abstract. Earth System Models (ESMs) participating in the Coupled Model Intercomparison Project Phase 5 (CMIP5) showed large uncertainties in simulating atmospheric  $CO_2$  concentrations. We utilize the Earth System Model Evaluation Tool (ESMValTool) to evaluate emission driven CMIP5 and CMIP6 simulations with satellite data of column-average  $CO_2$  mole fractions (XCO<sub>2</sub>). XCO<sub>2</sub> time series show a large spread among the model ensembles both in CMIP5 and CMIP6. Compared

- to the satellite observations, the models have a bias of +25 to -20 ppmv in CMIP5 and +20 ppmv to -15 ppmv in CMIP6, with the multi-model mean biases at +10 ppmv and +2 ppmv respectively. The derived mean atmospheric XCO<sub>2</sub> growth rate (GR) of 2.0 ppmv yr<sup>-1</sup> is overestimated by 0.4 ppmv yr<sup>-1</sup> in CMIP5 and 0.3 ppmv yr<sup>-1</sup> in CMIP6 for the multi-model mean, with a good reproduction of the interannual variability. All models capture the expected increase of the seasonal cycle amplitude (SCA) with increasing latitude, but most models underestimate the SCA. Any SCA derived from data with missing values can
- only be considered an "effective" SCA, as the missing values could occur at the peaks or troughs. The satellite data are a combined data product covering the period 2003–2014 based on the SCIAMACHY/ENVISAT (2003–2012) and TANSO-FTS/GOSAT (2009–2014) instruments. While the combined satellite product shows a strong negative trend of decreasing effective SCA with increasing XCO<sub>2</sub> in the northern midlatitudes, both CMIP ensembles instead show a non-significant positive trend in the multi-model mean. The negative trend is reproduced by the models when sampling them as the
- observations, attributing it to sampling characteristics. Applying a mask of the mean data coverage of each satellite to the models, the effective SCA is higher for the SCIAMACHY/ENVISAT mask than when using the TANSO-FTS/GOSAT mask. This induces an artificial negative trend when using observational sampling over the full period, as SCIAMACHY/ENVISAT covers the early period until 2012, with TANSO-FTS/GOSAT measurements starting in 2009. Overall, the CMIP6 ensemble shows better agreement with the satellite data than the CMIP5 ensemble in all considered quantities (XCO<sub>2</sub>, GR, SCA and
- trend in SCA). This study shows that the availability of column-integral  $CO_2$  from satellite provides a promising new way to evaluate the performance of Earth System Models on a global scale, complementing existing studies that are based on in situ measurements from single ground-based stations.

#### **1** Introduction

The Intergovernmental Panel on Climate Change (IPCC) Fifth Assessment Report (AR5) concluded that since 1950 many of

- the observed changes in the climate system are unprecedented in the instrument record, confirming an unequivocal warming of the climate system (IPCC, 2013). Increasing emissions of greenhouse gases (GHGs) are the key drivers of anthropogenic climate change. The most important anthropogenic greenhouse gas is carbon dioxide (CO<sub>2</sub>), with CO<sub>2</sub> emissions contributing more than half of the total global radiative forcing in 2011 relative to 1750 (IPCC, 2013). It is therefore important to monitor the long-term changes in atmospheric CO<sub>2</sub> concentrations, to understand the sources and sinks of carbon, and to provide reliable projections of future CO<sub>2</sub> concentrations under various scenarios.
- Photosynthesis causes a net uptake of atmospheric  $CO_2$  and thus declining atmospheric  $CO_2$  concentrations in the growing season. Conversely, atmospheric  $CO_2$  concentrations rise throughout the dormant season when there is a net release of  $CO_2$ from the land due to decomposition of organic matter in soils. This uptake and release of carbon by the terrestrial biosphere throughout the year causes a seasonal cycle of atmospheric  $CO_2$  (Keeling et al., 1989). The seasonal cycle amplitude (SCA)
- has been increasing over the last 50 years, with higher increases in higher latitudes (Barnes et al., 2016; Graven et al., 2013; Yin et al., 2018; Keeling et al., 1995; Keeling et al., 1996; Myneni et al., 1997; Piao et al., 2018). A number of studies have explored the effects of CO<sub>2</sub> fertilization, land-use change and climate warming on the SCA (Bastos et al., 2019; Zhao et al., 2016; Fernández-Martínez et al., 2019). Although models do not agree unanimously, the dominant effects are a positive trend in SCA due to the CO<sub>2</sub> fertilization combined with a negative trend due to climate warming. Some models however show a
- large positive trend due to climate warming (Zhao et al., 2016). Land-use is found to be a weaker effect in comparison to CO<sub>2</sub> fertilization and climate warming (Bastos et al., 2019; Fernández-Martínez et al., 2019). Most long-term measurements of CO<sub>2</sub> are from ground-based stations. In situ ground-based measurements at Mauna Loa (Hawaii, USA) started in 1958, providing the first evidence that fossil fuel combustion leads to a measurable increase in atmospheric CO<sub>2</sub> concentrations (Keeling et al., 1976). Other observatories around the globe now also measure atmospheric
- CO<sub>2</sub>, reporting an increase of about 45% since pre-industrial times (Ciais et al., 2013; Friedlingstein et al., 2019). Satellite measurements of CO<sub>2</sub>, with first near-infrared/short-wave-infrared (NIR/SWIR) nadir (downward-looking) based satellite retrievals starting in 2002, can complement the ground-based measurement network and provide regional and spatial distributions of CO<sub>2</sub>. The quantity obtained from measurements with NIR/SWIR satellite instruments is the column-average dry-air mole fraction of atmospheric CO<sub>2</sub>, denoted as XCO<sub>2</sub>. XCO<sub>2</sub> is a dimensionless quantity defined as the vertical column
- of CO<sub>2</sub> divided by the vertical column of dry air (i.e., all air molecules except water vapor) often given in ppmv (parts per million per volume). An analysis of growth rates (GR) and seasonal cycle amplitude (SCA) from satellite data and their sensitivity to growing season temperature anomaly presented in Schneising et al. (2014) shows a negative correlation between SCA and growing season temperature anomaly for the period 2003–2011, which was confirmed by Yin et al. (2018) for SCA anomaly in this timeframe. Satellite XCO<sub>2</sub> products are often used in combination with atmospheric transport inverse
- modelling approaches to obtain information on surface fluxes by using a global or regional transport model with free fit

parameters (Basu et al., 2013; Houweling et al., 2015; Reuter et al., 2014; Chevallier et al., 2014). The satellite data can also be used to constrain process parameters of a terrestrial biosphere model, e.g., as part of a Carbon Cycle Data Assimilation System (CCDAS, e.g. Kaminski et al. (2013)), and have been used for the evaluation of chemistry-climate models (Hayman et al., 2014; Shindell et al., 2013). In the last few years, satellite data have also been used in direct comparison to output from

70 climate models (e.g. Calle et al., 2019) characterizing rise and fall segments in seasonal cycles from GOSAT and comparing them to model output.

A large ensemble of climate model simulations for different type of experiments under common forcings is provided by the Coupled Model Intercomparison Project (CMIP), with output available for CMIP5 (Taylor et al., 2012) and more recently Phase 6 (CMIP6, Eyring et al. (2016a)). ESMs produce a large range in projected atmospheric CO<sub>2</sub>, as a result of uncertainties

- in the future evolution of natural fluxes (Arora et al., 2013; Friedlingstein et al., 2006). Overall CMIP5 models overestimate the carbon content of the atmosphere (Friedlingstein et al., 2014; Hoffman et al., 2014). The largest uncertainties are associated with the response of the land carbon cycle to changes in climate and atmospheric CO<sub>2</sub> (Friedlingstein et al., 2014; Hajima et al., 2014). The ability of ESMs to simulate the land and ocean contemporary carbon cycle has previously been investigated by Anav et al. (2013). They showed that most models were able to correctly reproduce the main climatic variables and their
- seasonal evolution, but found weaknesses in reproducing specific biogeochemical fields, such as a general overestimation leaf area index and photosynthesis. However, the magnitude of the global photosynthesis is not well constrained by observations, with estimates ranging between 112 and 169 PgC yr<sup>-1</sup> (Ryu et al., 2019), and the dataset used by Anav et al. (2013) is on the lower end of this range. For CMIP6, (Arora et al., 2020) analyzed simulations with a  $CO_2$  increase of 1 % per year to quantify the carbon-climate feedbacks. They found no significant change in behavior from CMIP5 to CMIP6, but lower absolute values

for models which included a nitrogen cycle. In this paper we focus on evaluating the growth rate and the seasonal cycle amplitude of simulated CO<sub>2</sub>, converted to XCO<sub>2</sub>, from CMIP ESMs which performed emission driven simulations with satellite observations in CMIP5 and CMIP6. The paper is structured as follows: the data products used in this study are introduced in Section 2. Section 3 describes the methods used, including the calculation of all derived quantities. A comparison between CO<sub>2</sub> flask measurements and XCO<sub>2</sub> measurements

at different locations is given in Section 4. The evaluation of CMIP simulations with satellite data is presented in Section 5, divided into sections focusing on the models' ability to simulate XCO<sub>2</sub> time series, growth rate and seasonal cycle amplitude. A summary and conclusion is given in Section 6.

#### 2 Data

#### 2.1 Observational datasets

#### 95 2.1.1 Satellite XCO<sub>2</sub>

We use the Observations for Model Intercomparisons Project (obs4MIPs) version 3 (O4Mv3) XCO<sub>2</sub> satellite data (Buchwitz et al., 2017a; Buchwitz et al., 2018). Obs4MIPs hosts observationally based datasets which have been formatted according to the CMIP model output requirements (e.g. variable definitions, coordinates, frequencies) in order to facilitate an easier comparison between observations and models (Ferraro et al., 2015; Teixeira et al., 2014; Waliser et al., 2020). The satellite

product used here is a gridded (Level 3) monthly data product with a 5° x 5° spatial resolution following the obs4MIPs format, produced as part of the Copernicus Climate Change Service (C3S). The O4Mv3 product is retrieved from the two satellite instruments SCIAMACHY/ENVISAT (Bovensmann et al., 1999; Burrows et al., 1995) and TANSO-FTS/GOSAT (Kuze et al., 2009).

This monthly mean XCO<sub>2</sub> satellite dataset covers a 14-year timespan (2003–2016). It is obtained by gridding the level 2 product

- (individual soundings) generated with the Ensemble Median Algorithm (EMMA, Reuter et al. (2013)), in this case EMMA version 3.0 (EMMAv3, Reuter et al. (2017)). EMMA combines several different XCO<sub>2</sub> level 2 satellite data products from SCIAMACHY/ENVISAT (2003–2012) and TANSO-FTS/GOSAT (2009–2016) and includes a bias correction to all products during overlap phases, resulting in a good agreement during the overlap period. This product was validated against Total Carbon Column Observing Network (TCCON, Wunch et al. (2011)) ground-based observations of XCO<sub>2</sub>, revealing a +0.23
- 110 ppmv global bias, a relative accuracy (defined as standard deviation of the station-to-station biases) of 0.3 ppmv, and a very good stability in terms of a linear bias trend ( $-0.02 \pm 0.04$  ppmv yr<sup>-1</sup>) (Buchwitz et al., 2017b). While the dataset ends in 2016, our evaluation only goes up to the year 2014 because the historical simulations for CMIP6 end in 2014 and scenarios from the emission-driven simulations that could be used to extend the runs are not yet available for all considered models.
- The number of observations depends significantly on the location with most points over locations with low cloud cover, high surface reflectivity and (at least) moderate to high sun elevation. Coverage over ocean is sparse as ocean retrievals are only included from GOSAT sun-glint mode observations - outside of glint conditions the reflectivity of water is very low in the NIR/SWIR spectral region. Figure 1 shows the mean monthly coverage of the dataset for 2003–2014. In Section 5 we will show that taking into account this sampling in the evaluation of ESMs is essential for a proper comparison.

The dataset also contains uncertainty estimates for each grid cell, with a mean value of 0.92 ppmv, accounting for both statistical uncertainties from the individual soundings and uncertainties from potential regional and temporal biases (Buchwitz

statistical uncertainties from the individual soundings and uncertainties from potential regional and temporal biases (Buchwitz et al., 2017a). However, the overall uncertainties are small compared to inter-model differences (see Section 3.1), and are therefore neglected in our analysis.

#### 2.1.2 Surface CO<sub>2</sub> measurements

For the comparison of satellite  $XCO_2$  and surface  $CO_2$  data in Section 4 we have obtained surface flask measurements from

the NOAA ESRL Carbon Cycle Cooperative Global Air Sampling Network (Dlugokencky et al., 2020). Measurement sites at locations with no available satellite data were excluded from the analysis, which ruled out the four baseline observatories in Mauna Loa, Samoa, as well as the South Pole and Point Barrow sites. Furthermore, sites which did not collect data during the period from 2003–2014 were discarded. From the remaining sites, a sample of five sites was chosen which had the best coverage of different latitudes, and when latitudes were similar, different longitudes were selected for increased spatial coverage. The selected sites are listed in Table 1.

#### 2.2 Model simulations

We use monthly mean output data from ten CMIP5 and ten CMIP6 models which performed emission driven simulations, with three of the CMIP5 and five of the CMIP6 models including a nitrogen cycle. Tables 2 and 3 list all the CMIP5 and CMIP6 models used in this paper along with their atmosphere, land and ocean model component, respectively. Only models

- with interactive carbon cycle are able to perform the emission driven simulations, in which the emissions rather than the concentrations of the greenhouse gases are prescribed (Taylor et al., 2012; Eyring et al., 2016a). This allows the carbon cycle in the models to react to changes in climate and atmospheric CO<sub>2</sub>, by adjusting their carbon fluxes to the new climate conditions and providing the atmospheric CO<sub>2</sub> concentration as an output (Friedlingstein et al., 2014). In order to facilitate the comparison between the satellite data and the CMIP5 emission driven simulations, the historical simulations (1850–2005) were extended
- beyond 2005 with simulations from the Representative Concentration Pathway (RCP) 8.5 (2006–2100), for which most ESM simulations are available. Since the period of observations only extends a decade beyond the historical runs, the choice of emissions scenario has a negligible impact on the results that we present below. For CMIP6 only the historical simulations are used, which end in 2014. For CMIP5, only one model had more than one ensemble member performing the emission driven RCP 8.5 simulation and thus only one ensemble member for each model has been used. In CMIP6, several models have three
- or more ensemble members. We consider all of them in Figure 3 for the timeseries to show the models' intrinsic variability, but then proceed the analysis with only the first ensemble member for each model. The different initial value ensemble members similarly to each other for the analysis presented in this paper, and using an ensemble mean would reduce the interannual variability found in each individual member.

#### 3 Methods

### 150 **3.1 Sampling of observations and models**

For comparison of model and satellite data, first the  $CO_2$  data of the models were converted to  $XCO_2$  data. The model data was interpolated to the grid of the satellite dataset using a bilinear interpolation scheme and grid cells with missing values in

the satellite data were also set to missing values in the model fields. Further sampling considerations are discussed in more detail in Section 5.3.2 and in Appendix A.

Most analysis is carried out with regional averages covering several grid cells. Unless specifically stated otherwise, these are calculated by taking the arithmetic averages over all grid cells weighted by their area for each month.

#### 3.2. Calculation of growth rate, seasonal cycle amplitude and growing season temperature anomaly

We compute the growth rate (GR) following the method described in Buchwitz et al. (2018). Monthly resolved annual growth rates are calculated by subtracting the  $XCO_2$  value 6 months in the future from the one 6 months in the past. Then these monthly

resolved growth rates are averaged to a yearly GR for a calendar year, and any years with less than 7 months of data are flagged as missing. The addition of the 7–month data availability was introduced to be consistent with the constraint on SCA as explained below.

We define the SCA as the peak-to-trough amplitude in a calendar year of the detrended time series. The time series is detrended with the cumulative sum of monthly growth rates, using the annual mean growth rates as substitution for missing values where

- necessary. The SCA is calculated by subtracting the minimum from the maximum value for each year with a minimum data availability of 7 months. When investigating the seasonal cycle of observationally sampled simulations at higher latitudes, the maximum value of the time series was generally only accounted for if more than 7 months of data were available. We therefore introduce the cutoff of 7–month data availability to preserve as many peaks as possible without restricting the data too much. However, as peak preservation cannot be guaranteed when any missing values are present, we can only speak of an effective
- SCA. The absolute SCA is not as important in our comparison, because we use the same sampling for both the model and observations.

The growing season temperature anomaly  $\Delta T$  is calculated from the GIStemp (Hansen et al., 2010) temperature anomaly map following Schneising et al. (2014). The data is masked to include only vegetated areas, using the MODIS Land Cover Classification (Friedl et al., 2010; Channan et al., 2014). Surface temperature anomalies are calculated with respect to their

monthly climatologies. The data is averaged over the growing season if it covers only one hemisphere (AprilSeptember for the Northern Hemisphere, December to May for the Southern Hemisphere). Additionally, if the data covers both hemispheres, the whole year is taken into account. The growing season averages are taken because the temperature has a large influence on the plant growth and the resulting biospheric CO<sub>2</sub> fluxes, which in turn drive both the SCA and interannual variability of the GR (Schneising et al., 2014).

#### 180 **3.3 Earth System Model Evaluation Tool (ESMValTool)**

All figures in this paper were produced with the Earth System Model Evalution Tool (ESMValTool) version 2.0 (v2.0) (Righi et al., 2020; Eyring et al., 2020; Lauer et al., 2020). Since its first release in 2016 (Eyring et al., 2016b) the ESMValTool has been further advanced facilitating analysis of many different ESM components, providing well-documented source code and scientific background of implemented diagnostics and metrics and allowing for traceability and reproducibility of results

- (provenance). ESMValTool v2.0 has been developed as a large community effort to specifically target the increased data volume of CMIP6 and the related challenges posed by analysis and evaluation of output from multiple high-resolution and complex ESMs. For this, the core functionalities have been completely rewritten in order to take advantage of state-of-the-art computational libraries and methods to allow for efficient and user-friendly data processing (Righi et al., 2020). Common operations on the input data such as regridding or computation of multi-model statistics are now centralized in a highly
- optimized preprocessor written in Python. The ESMValTool v2.0 includes an extended set of large-scale diagnostics for quasioperational and comprehensive evaluation of ESMs (Eyring et al., 2020), new diagnostics for extreme events, regional model and impact evaluation and analysis of ESMs (Weigel et al., 2020, submitted), as well as diagnostics for emergent constraints and analysis of future projections from ESMs (Lauer et al., 2020). For the study here, a new ESMValTool recipe has been developed that can be used to reproduce all plots of this paper.

#### 195 4. Comparison of XCO<sub>2</sub> and surface CO<sub>2</sub>

Until recent years, most model-observation comparisons have been carried-out using in situ surface  $CO_2$  data (e.g. Wenzel et al., 2016). As such, it is interesting to compare the differences between  $XCO_2$  and surface  $CO_2$  at different locations. Figure 2 shows a comparison of time series between both kinds of observations and the multi-model mean for both  $XCO_2$  and surface  $CO_2$  for CMIP6 (top) and CMIP5 (bottom) models. The multi-model mean for both  $XCO_2$  and surface  $CO_2$  is offset to have

- the same mean value as the satellite data, and this offset is noted above each time series panel. It is interesting to note that the offset appears to be larger at higher latitudes thus showing a different latitudinal gradient between the models and the satellite data, indicating potential issues with surface fluxes or transport in the models. The multi-model mean and satellite data are averaged between all grid cells covering a  $5^{\circ} \times 5^{\circ}$  radius around the stations, which results in a maximum of four grid cells to be considered. The mean and growth rate of XCO<sub>2</sub> and surface CO<sub>2</sub> are in very good agreement, while the multi-model mean
- overestimates both variables at all sites, with the overestimated mean XCO<sub>2</sub> arising from the effect of higher growth rates over time. Furthermore, the offset from the modelled surface CO<sub>2</sub> is higher than that of XCO<sub>2</sub>, while this difference is smaller for CMIP5. This might be due to the fact that the CMIP5 offset for multi-model mean XCO<sub>2</sub> was larger overall with approximately 10 ppmv compared to the CMIP6 offset of approximately 2 ppmv.
- SCA is higher at higher latitudes, and also generally higher at the surface compared to the column average. This is to be expected as the processes dominating the seasonal cycle, respiration and photosynthesis, take place at the surface leading to the higher SCA for station data and surface CO<sub>2</sub> from models compared to the XCO<sub>2</sub> values. Mixing of air coming from lower latitudes with lower SCA dampens the SCA in the column compared to surface SCA. This is evident in the increasing SCA difference between XCO<sub>2</sub> and surface CO<sub>2</sub> going from low latitude to high latitude stations, with no discernible seasonal cycle in the southern hemisphere due to the lack of land and vegetation. The multi-model mean follows this trend in the observations,
- although it underestimates the higher latitude SCA, with a larger underestimation at the surface while capturing the XCO<sub>2</sub>

SCA relatively well. As this study aims at evaluating model simulations with satellite data, further analysis is restricted to XCO<sub>2</sub>.

#### 5. Evaluation of CMIP simulations with satellite data

#### 5.1. XCO<sub>2</sub> time series

- The globally averaged time series of XCO<sub>2</sub> is shown in Figure 3 on the top panel, with CMIP6 (a) and CMIP5 (b) models sampled as the satellite observations (see Section 3.1). The observational uncertainty is too small to be seen in this plot and is therefore neglected. The middle panel shows the monthly resolved annual growth rate and the bottom panel the detrended seasonal cycle. All available ensemble members for CMIP6 models are used to show their internal variability. All ensemble members perform similar to one another. The multi-model mean is computed using the first ensemble member, which is also used in the further analysis. As in Figure 2, an increase of XCO<sub>2</sub> with time and a pronounced seasonal cycle for all models can
- used in the further analysis. As in Figure 2, an increase of XCO<sub>2</sub> with time and a pronounced seasonal cycle for all models can be seen. The focus here is on the absolute values, as the trend (GR) and SCA are discussed in dedicated sections below. The CMIP6 models display a large range of absolute XCO<sub>2</sub> values, ranging from an underestimation by 15 ppmv (MRI-ESM2.0 and MIROC-ES2L) to an overestimation by 20 ppmv (GFDL-ESM4). The models closest to the observations is CNRM-ESM2-1 which reproduces the mean value well, with the next closest models being NorESM2-LM and MPI-ESM1.2-LR both
- overestimating XCO<sub>2</sub> by about 5 ppmv. The multi-model mean shows an overestimation by approximately 2 ppmv or equivalently a time-shift of 1 year. While the spread in the models has not decreased much since CMIP5, the overestimation of the multi-model mean has decreased from 10 ppmv to 2 ppmv. Furthermore, CMIP6 models which have predecessors in CMIP5 show similar biases as their predecessors, besides the MIROC models, which overestimated the mean by 15ppmv in CMIP5 and underestimates it by that much in CMIP6. Both MRI models underestimate XCO<sub>2</sub> significantly, while GFDL-
- ESM4 overestimates the atmospheric content even more. The MRI-ESM1 model was the only model in CMIP5 to underestimate XCO<sub>2</sub> with respect to the observations, and this by about 20 ppmv. This model underestimates the historical warming, causing plant and soil respiration to be too low, which leads to a larger land sink and a reduced atmospheric CO<sub>2</sub> concentration (Adachi et al., 2013). This underestimation has been reduced by about 5 ppmv in CMIP6. The GFDL models show an overestimation of about 15 ppmv in both ensembles, and both CanESM models are 10 ppmv too high. A minor
- improvement can be seen for NorESM-LM over NorESM1-ME, with a decrease of the overestimation from 15 to 10 ppmv.

#### 5.2 Growth Rate

The middle panel of Figure 3 shows that while models capture the interannual variability of the growth rate quite well, they overestimate the mean growth rate compared to the observations. The correlation coefficient for the multi-model mean is at 0.48 in CMIP6 and 0.07 in CMIP5 which shows a large improvement in this area. The pronounced feature in 2009 is due to

the introduction of the GOSAT data which changed the shape of the seasonal cycle and thus due to its calculation the monthly resolved annual growth rate. Fortunately, this feature gets averaged out when computing the annual growth rate and does not

tangibly affect our conclusions. Figure 4 shows the global mean GR of  $XCO_2$  for 2003–2014 and its standard deviation over all years depicted as error bars, with the observations shown in black and the multi-model mean in red. The annual mean GR of the satellite data is  $1.9 \pm 0.4$  ppmv yr<sup>-1</sup>, while the CMIP5 models (right) range from  $1.5 \pm 0.4$  ppmv yr<sup>-1</sup> (MRI-ESM1) to 3.0

- $\pm 0.9$  ppmv yr<sup>-1</sup> (CanESM2) with a multi-model mean of 2.4  $\pm 0.4$  ppmv yr<sup>-1</sup>. In CMIP6 (left), the multi-model mean is slightly lower at 2.3  $\pm 0.3$  ppmv yr<sup>-1</sup> and the spread has decreased by 0.6 ppmv yr<sup>-1</sup>, with a range from 1.7  $\pm 0.4$  ppmv yr<sup>-1</sup> (MRI-ESM2.0) to 2.6  $\pm 0.7$  ppmv yr<sup>-1</sup> (GFDL-ESM4). As expected from Figure 3, the models - with the exception of MRI-ESM1, MRI-ESM2.0 and MIROC-ES2L - overestimate the growth rate, leading to an increased XCO<sub>2</sub> level in the present-day atmosphere compared to observations. The interannual variability of the growth rate for the models is generally higher than 255 that of the observations, but well reproduced in the multi-model mean.
  - Emergent Constraints are relationships defined using an ensemble of models, between a measurable aspect of current or past climate and an aspect of Earth system feedback in the future, which can be constrained using observational data (Eyring et al., 2019). In Appendix C we have tried to reproduce the trend in interannual variability (IAV) of CO<sub>2</sub> growth rate to IAV of tropical temperature used by Cox et al. (2013) to develop an emergent constraint on the sensitivity of tropical land carbon to
- climate change, but were unable to find a significant trend for this much shorter satellite-derived time series. The spatial variability of the GR is small, as CO<sub>2</sub> is long-lived and well mixed in the atmosphere with a one year mean interhemispheric crossing time. Thus there should be no significant regional changes on an annual level. Buchwitz et al. (2018) found the growth rate of the satellite dataset to be in agreement with those quoted by NOAA ESRLs global and Mauna Loa time series, as well as robust over several latitude bands. Our own analysis also shows only very small regional differences in
- the growth rate (not shown). No significant changes to the annual growth rates due to the satellite spatial coverage were found.

#### 5.3 Seasonal Cycle Amplitude

This section about the seasonal cycle amplitude (SCA) is divided into two subsections, with the first one taking a closer look at inter-model differences, while the second subsection is devoted to the impact of observational sampling.

#### 5.3.1 Model differences

The lower panel in Figure 3 shows the detrended global seasonal cycle for all models. Models in CMIP6 (a) show a strong improvement in their ability to capture both the seasonal cycle amplitude, as well as its phase compared to CMIP5 (b), but still underestimate the SCA. The correlation coefficient to the observed seasonal cycle is 0.93 in CMIP5 and 0.98 in CMIP6 for the multi-model mean. The only model in CMIP6 to significantly underestimate the seasonal cycle amplitude is CNRM-ESM2-

1. Two errors have been identified causing this dampened seasonal cycle: Ocean carbon fluxes are lagged in time, and in the

emission driven simulations, CO<sub>2</sub> is considered as an active tracer and coupled with atmospheric chemistry. These chemical fields are restored to global mean climatological concentrations at the model surface, acting as a damping component to the CO<sub>2</sub> concentrations (Séférian et al., 2019). Figure 5 shows maps of the climatological mean SCA (2003–2014) for all models, with the global mean given in the top right and the zonal averages shown in the panel attached to the right of the maps. All

CMIP6 models (Figure 5a) underestimate the SCA compared to satellite observations (Figure 6 middle) in the global mean,

with the closest mean SCA being MIROC-ES2L. This underestimation was already present in CMIP5 (Figure 5b), with several studies discussing it for surface CO<sub>2</sub> SCA (Wenzel et al., 2016; Graven et al., 2013). In CMIP6 the multi-model mean has a globally averaged mean SCA of 3.25 ppmv, compared to 2.92 ppmv for CMIP5, while the observations show an effective SCA of 5.89 ppmv (Figure 6).

Both models and observations show the well-known increasing SCA with increasing latitude, due to the more pronounced

- seasonal cycle of the climate at higher latitudes. Most models show a decreased growth from 0-30°N, with higher SCA increases in the lower tropics and northern midlatitudes. Overall the zonal distribution is quite similar throughout the models, with UK-ESM1-0-LL showing increased SCA at 30-90°S. Tropical land regions in northern South America, Africa and south east Asia show increased SCA values compared to the ocean SCA at this latitude for the same model. While in the GFDL CMIP5 models this was so pronounced that these regions showed a higher SCA than the higher latitudes (Dunne et al., 2013),
- this is no longer the case for GFDL-ESM4 in CMIP6. Dunne et al. (2013) attributed the GFDL problem in CMIP5 to the seasonality of respiration in the northern latitudes and an Amazonian low-precipitation bias which reversed the seasonal cycle synchronizing it with the African and Oceanian rain forests. The improvement in CMIP6 is due to a reduced ocean-atmosphere CO<sub>2</sub> flux in the Southern Hemisphere, as well as the reduction of the high tropical land-atmosphere fluxes expressed over the ocean (Dunne et al., 2020).
- The SCA in the CMIP5 MPI-ESM-LR model is on average twice as large as the observed one. The high SCA has been discussed in Giorgetta et al. (2013) where it was attributed to a combination of an overestimation of net primary productivity in ocean and land biology and uncertainties in atmospheric tracer transport. A particularly severe overestimation was seen in the Southern Hemisphere when comparing to station data. As shown in Figure 6, we additionally find a large overestimation in XCO<sub>2</sub> SCA in the Northern Hemisphere, in particular in the extra-tropics. For the CMIP6 successor model, MPI-ESM1.2-
- LR, the SCA is still the highest of the model ensemble, but is no longer twice as high as the other models. However, it now shows a more pronounced SCA over the tropical land regions mentioned above, which was not as dominant in CMIP5. It is known that nitrogen limitations tend to suppress CO<sub>2</sub> fertilization (Reich et al., 2006). Of the four models with the lowest overall SCA in CMIP5 (CESM1-BGC, FIO-ESM, NorESM1-ME and BNU-ESM), three of them BNU-ESM, CESM1-BGC, NorESM1-ME include a nitrogen cycle. The SCA of NorESM1-ME and CESM1-BGC are very similar, which can be
- attributed to sharing the same land model (CLM4). FIO-ESM uses the predecessor CLM3.5 and is also comparable to the other two models. It was found that CLM4 had an unrealistically strong nitrogen limitation, which has been reduced in CLM5 (Wieder et al., 2019). In CMIP6, ACCESS-ESM1.5, MPI-ESM1.2-LR, MIROC-ES2L, NorESM2-LM and UKESM1.0-LL include a nitrogen model but none of them share the same land model. While ACCESS-ESM1.5 has the second lowest overall SCA, MPI-ESM1.2-LR and MIROC-ES2L have the highest, and thus the observation from CMIP5 models that N-cycle models
- feature a lower SCA no longer stands for CMIP6.

#### **5.3.2 Influence of Sampling**

There are a number of ways to compare model SCA to observational SCA, beginning with a grid box comparison. Figure 6 shows a comparison of the multi-model mean of CMIP6 (Figure 6a) and CMIP5 (Figure 6b) to observations. The top right shows the unsampled SCA. The top left panel shows the effective SCA when using observational sampling and the middle

- panel the satellite data's effective SCA. All numbers are given in ppmv. For an easier comparison the bottom panels show the absolute difference plots, with the left panel depicting the difference between sampled model and observations, and the right panel the difference between the sampled and unsampled model. Observational sampling slightly lowers the SCA, which is to be expected, as it could lead to masking out the peaks or troughs. While this effect was minimized by imposing the restriction of only computing the SCA of a year when at least 7 months of data are available, it is still a possibility. We therefore classify
- this SCA as an "effective SCA". However, the SCA does not seem to decrease significantly through sampling and the difference does not follow a trend in latitude, so a grid box comparison seems feasible. This paves the way for more comprehensive spatial investigations, which previously relied on data from ground-based stations with sparse spatial coverage. While the stations provide data in higher latitudes that the satellite dataset does not cover, in the tropics and mid-latitudes the spatial coverage of the satellite is superior to the ground-based stations. A downside with this approach is the sparsity of the
- data when using observational sampling. Furthermore, this becomes a computationally expensive operation, as the SCA will need to be calculated for each grid box.

Another approach often used in model analysis is area averaging, e.g. over different latitude bands like the tropics or the northern mid-latitudes. Using surface flask measurements Wenzel et al. (2016) found an increased SCA with rising  $CO_2$  concentration for CMIP5 using model data from the full historical simulation 1850 to  $2005 - CO_2$  fertilization -, and used this

- to establish an emergent constraint on the fertilization of terrestrial gross primary productivity (GPP). Figure 7 shows the SCA trend of CMIP6 models versus XCO<sub>2</sub> for 2003–2014 in the northern mid-latitudes (30–60° N), including a linear regression including slope, mean SCA, Pearson correlation coefficient and p-value. The left panel shows the trend in the unsampled models, while the right one shows the trend when following the sampling of the satellite data. The SCA was computed after a weighted area-average on the XCO<sub>2</sub> time series. While some of the unsampled models show an increasing SCA trend with
- increasing XCO<sub>2</sub>, which is in agreement with the findings from Wenzel et al. (2016), others don't show a statistically significant trend and the multi-model mean shows an insignificant positive trend. The sampled model data (right) show a significant negative trend. Calculating the average with a zonal average before summing up the latitude bands does not change this result.

To investigate this change in trend due to observational sampling, Figure 8 shows the data coverage for different time periods,

2003–2008 for SCIAMACHY only measurements (top), the overlap between the two satellites in 2009–2012 (middle), and 2013–2014 for the GOSAT satellite only (bottom), with the pattern marking areas with a coverage of 50 % or above. Above 50° N SCIAMACHY measurements include more areas with 50 % or more coverage compared to GOSAT measurements. With a larger SCA in higher latitudes, it implies that SCIAMACHY would have a larger average SCA over this region

compared to GOSAT, hence artificially generating a decreasing trend in the observed SCA, when moving from SCIAMACHY

- to GOSAT. Figure 9 shows the CMIP6 effective SCA trend with XCO<sub>2</sub> using SCIAMACHY and GOSAT masks obtained from masking out points with less than 50 % coverage. While the slopes remain largely the same, the mean effective SCA is higher in the models using the SCIAMACHY mask than when using the GOSAT mask. This mean SCA difference is larger than the total SCA difference within a model using the same sampling for the whole time period. Thus when considering the observational time series and its full sampling, the trend intrinsic to the model is dominated by the negative SCA difference
- going from the SCIAMACHY to the GOSAT data coverage and thus changed to the negative trend seen in the observations. We can therefore attribute at least part of the negative trend in the satellite data to the different data coverage of the two satellites. We are able to reproduce this negative trend with the models, when these are sampled consistently with the satellite data. This study on sampling also holds true for CMIP5 models, with the equivalent figures shown in Appendix B (Figures B1 and B2).
- Further impacts on  $CO_2$  concentrations could come through temporal sampling, such as the fact that the satellites data only includes measurements with low cloud cover and is limited to 13:00 local time. While cloud cover can impact photosynthesis, the response can be fundamentally different for various ecosystems (Still et al., 2009), we expect a larger effect from the diurnal cycle in  $CO_2$  which is included in the model means but not the satellite data. Due to a lack of  $CO_2$  data from models with a higher temporal resolution, this effect cannot be estimated in this study.

#### 360 6. Summary and Conclusion

In this paper we have evaluated the performance of CMIP5 and CMIP6 ESMs with interactive carbon cycle (Tables 2 and 3) against column integral CO<sub>2</sub> (XCO<sub>2</sub>) data from satellite retrievals. Our analysis has compared ESM simulations to the 2003-2014 Obs4MIPs XCO<sub>2</sub> satellite dataset O4Mv3 retrieved from radiance spectra measured by the SCIAMACHY/ENVISAT (20032012) and TANSO-FTS/GOSAT (2009–2014) satellite instruments. The O4Mv3 data product has a spatial resolution of 5°x5° and monthly time resolution. For CMIP5, the historical simulations covering the period 2003–2005 were combined with simulations from the RCP 8.5 scenario (2006–2014) and for CMIP6 the historical simulations were used (2003–2014). The evaluation of the CMIP models with the satellite data focused on the time series, the growth-rate (GR) and the seasonal cycle amplitude (SCA) XCO<sub>2</sub>. All SCAs computed with a masked time series are considered to be "effective" SCAs due to the possibility of masking out peaks and troughs.

The XCO<sub>2</sub> time series comparison shows that most models overestimate the carbon content of the atmosphere relative to the satellite observations in both model ensembles, with a lower overestimation for the CMIP6 models of 2 ppmv for the multimodel mean and a wide range of individual model differences of -15 ppmv to +20 ppmv. The CMIP5 models overestimate by 5 to 25 ppmv with the exception of the MRI-ESM1 model, which underestimates by 20 ppmv. The CMIP5 multi-model mean overestimates by 10 ppmv compared to the observations, which has also previously been found for surface comparisons 375 (Friedlingstein et al., 2014;Hoffman et al., 2014). Overall, CMIP6 models follow the same trends as their CMIP5 counterparts, but with reduced systematic biases.

The XCO<sub>2</sub> annual mean growth rate is typically slightly overestimated compared to the observational value of  $1.9 \pm 0.4$  ppmv yr<sup>-1</sup>. CMIP6 models range from  $1.7 \pm 0.4$  ppmv yr<sup>-1</sup> (MRI-ESM2.0) to  $2.6 \pm 0.7$  ppmv yr<sup>-1</sup> (GFDL-ESM4) with a multi-model mean of  $2.3 \pm 0.3$  ppmv yr<sup>-1</sup>. CMIP5 models have a slightly higher multi-model mean growth rate of  $2.4 \pm 0.4$  ppmv yr<sup>-1</sup>, and

a larger spread, with the CMIP5 lowest model being MRI-ESM1 at  $1.5 \pm 0.4$  ppmv yr<sup>-1</sup> and the highest CMIP5 growth rate shown by CanESM2 at  $3.0 \pm 0.9$  ppmv yr<sup>-1</sup>.

All models capture the expected increase of the SCA with increasing latitudes, but most models underestimate the SCA to differing degrees in different regions. This result is in line with previous studies (Wenzel et al., 2016;Graven et al., 2013). Models with similar model components show similar behavior, with models including a nitrogen cycle generally showing a

- lower SCA in CMIP5, but this influence is not clear in CMIP6. Finally, the connection between SCA and XCO<sub>2</sub> was investigated in the northern midlatitudes. Most models from both ensembles show a positive trend, i.e., an increase of the SCA with XCO<sub>2</sub>, consistent with findings for surface CO<sub>2</sub> (Wenzel et al., 2016). However, the satellite product shows a strong negative trend in contrast to the models and surface based observations. We have attributed this trend reversal to the sampling characteristics of the satellite products. The average effective SCA is higher for models sampled according to the
- SCIAMACHY/ENVISAT as opposed to the TANSO-FTS/GOSAT mean data coverage. As the early time series is based solely on the SCIAMACHY/ENVISAT data and the last years only use data from TANSO-FTS/GOSAT, this introduces an artificial negative trend which dominates the positive trend shown by the unsampled models. This demonstrates the importance of equal sampling of models and observations in model evaluation studies.
- There are several ways to improve on this analysis in the future. With more available future scenario simulations, the analysis can be extended to a longer time series, making use of longer observational timeseries, such as the one introduced in Reuter et al. (2020). Higher temporal resolution of the models would enable studies on the effect of the diurnal cycle of  $CO_2$  on the monthly mean and also allow for the construction of a co-located time series with the Level 2 satellite data. This could help highlight some of the causes of model biases by being able to pinpoint time and space where they occur more precisely. Model biases may also result from the CMIP experimental design, such as requiring the climate state to be in equilibrium in 1850
- while the real world may not have been (Bronselaer et al., 2017), or the parametrizations of biological and physical processes not allowing the system to change rapidly enough (Hoffman et al., 2014). Along with a longer time series, newer satellites, such as OCO-2 or the planned Sentinel 7 bring higher resolutions and more data, potentially helping to fill the gaps and reduce the impact of the sampling we discussed in Section 5.3.2.

Overall, the CMIP6 ensemble shows an improved agreement with the satellite data in all considered quantities (mean XCO<sub>2</sub>,

growth rate, SCA and trend in SCA), with the biggest improvement shown in the mean  $XCO_2$  content of the atmosphere. The paper demonstrates the great potential of satellite data for climate model evaluation as it allows to go beyond regional means or single point observations from in situ data, and also enables the investigation of regional effects on SCA, such as the increase in SCA at higher latitudes.

#### 410 Appendix A. Calculation of XCO<sub>2</sub>

Here we documents the general procedure used to compute model  $XCO_2$  for comparison with the satellite-based obs4MIPs product following the description in (Buchwitz and Reuter, 2016).

$$XCO_2 = \frac{\sum n_d \cdot c_{CO_2}}{\sum n_d} \tag{A1}$$

Here,  $c_{CO_2}$  represents the modeled CO<sub>2</sub> dry air mole fraction on model layers (i.e., layer centers or full levels) and  $n_d$  the 415 number of dry air particles (air molecules excluding water vapor) within these levels. The summations are performed over all model layers. The number of dry air particles can be computed as follows:

$$n_d = \frac{N_a \cdot \Delta p \cdot (1-q)}{m_d \cdot g} \tag{A2}$$

 $N_a$  is the Avogadro constant (6.022140857  $\cdot 10^{23} mol^{-1}$ ) and  $m_d$  the molar mass of dry air (28.9644  $\cdot 10^{-3} kg mol^{-1}$ ).  $\Delta p$  is the pressure difference (in hPa) computed from the model's pressure levels (i.e., layer boundaries or half levels) surrounding the model layers, q is the modeled specific humidity (in kg/kg), and q the gravitational accelaration approximated by:

$$g = \sqrt{g_0^2 - 2 \cdot f \cdot \phi} \tag{A3}$$

This includes the model's geopotential  $\phi$  (in m<sup>2</sup>s<sup>-2</sup>) on layers, the free air correction constant  $f = 3.0825959 \cdot 10^{-6}s^{-2}$ , and the gravitational acceleration  $g_0$  on the geoid approximated by the international gravity formula depending only on the latitude  $\varphi$ :

$$g_0 = 9.780327 \cdot [1 + 0.0053024 \cdot \sin^2(\varphi) - 0.0000058 \cdot \sin^2(2\varphi)]$$
(A4)

#### Appendix B. Satellite data mean coverage impact on CMIP5 SCA trend

The analysis from Section 5.3.2. on the influence of the satellite data mean coverage on the trend of the SCA was also performed for CMIP5. Figures B1 and B2 are the CMIP5 equivalent of Figures 8 and 10. The CMIP5 models support the analysis of the CMIP6 models and show that the different satellite data coverage results in a different mean effective SCA,
with a higher mean effective SCA for SCIAMACHY (2003–2012) than GOSAT (2009–2014) mean data coverage, which

overshadows the positive trend and causes it to flip to a negative one in most models.

#### Appendix C. Trend of growing season temperature and interannual variability of CO<sub>2</sub> growth rate

Cox et al. (2013) developed an emergent constraint on the sensitivity of tropical land carbon to climate change using the sensitivity of the interannual variability (IAV) of  $CO_2$  growth rate to the IAV of tropical temperature, which was later adapted

- by Wenzel et al. (2014) to CMIP5 models. Figure C1 shows the sensitivity of the IAV of  $XCO_2$  growth rate to the tropical growing season temperature IAV for CMIP6 (left) and CMIP5 (right) models, both compared with observations. The observational temperature is taken from the GISTEMP v4 dataset (Lenssen et al., 2019) and the models use their own modeled temperature. We find an observational value of  $-0.23 \pm 0.70$  ppmv yr<sup>-1</sup> K<sup>-1</sup> for the 2003-2014 period. However, when using the full span of the satellite data until 2016, the slope increases to  $0.75 \pm 0.6$  ppmv yr<sup>-1</sup> K<sup>-1</sup> (not shown), as the additional years
- show both a high growing season temperature and GR IAV, coinciding with a strong El Nino. This shows that the time period 2003–2014 is not sufficiently long to reproduce the emergent constraint, although this may become feasible once CMIP6 emission driven future simulations are available for a longer time overlap between models and observations. CMIP5 model values for the timeframe 2003–2014 range from 0.53 ± 0.51 ppmv yr<sup>-1</sup> K<sup>-1</sup> (NorESM1-ME) to 3.14 ± 0.63 ppmv yr<sup>-1</sup> K<sup>-1</sup> (MRI-ESM1), with only CESM1-BGC showing a negative trend of -0.64 ± 0.55 ppmv yr<sup>-1</sup> K<sup>-1</sup>. The multi-model mean has a value of 1.79 ± 0.80 ppmv yr<sup>-1</sup> K<sup>-1</sup>. In CMIP6 the range is significantly decreased with a minimum of 1.14 ± 0.56 ppmv yr<sup>-1</sup> K<sup>-1</sup>
- (ACCESS-ESM1.5) to a maximum of  $3.37 \pm 0.71$  ppmv yr<sup>-1</sup> K<sup>-1</sup> (CanESM5-CanOE), and a multi-model mean of  $1.14 \pm 0.37$  ppmv yr<sup>-1</sup> K<sup>-1</sup>.

#### 7. Data and Code Availability

The O4Mv3 XCO<sub>2</sub> data product is available via the Copernicus Climate Change Service (C3S, https://climate.copernicus.eu/)

- Climate Data Store (CDS) (https://cds.climate.copernicus.eu/), accessed Aug 2018. The surface flasks measurements were obtained online (ftp://aftp.cmdl.noaa.gov/data/trace gases/co2/flask/surface/) from Dlugokencky et al. (2018), accessed Aug-2018. Surface temperature anomalies were obtained from the GISTEMP Team, 2020: GISS Surface Temperature Analysis (GISTEMP), version 4. NASA Goddard Institute for Space Studies. Dataset accessed 2020-02-13 at data.giss.nasa.gov/gistemp/. The MODIS IGBP Land Cover Type Classification was obtained from 455 http://glcf.umd.edu/data/lc/ (accessed: 2018-01-31). CMIP data is available on various ESGF nodes (e.g. https://esgf-
  - <u>data.dkrz.de/search/cmip5-dkrz/</u>) (Williams et al., 2009), with CMIP6 data DOIs given in Table 3.
    ESMValTool v2.0 (Eyring et al., 2020; Lauer et al., 2020; Righi et al., 2020) is released under the Apache License, VERSION 2.0. The latest release of ESMValTool v2 is publicly available on Zenodo at <u>https://doi.org/10.5281/zenodo.3401363</u>. The source code of the ESMValCore package, which is installed as a dependency of the ESMValTool v2, is also publicly available
- on Zenodo at <u>https://doi.org/10.5281/zenodo.3387139</u>. ESMValTool and ESMValCore are developed on the GitHub repositories available at <u>https://github.com/ESMValGroup</u>. The corresponding recipe that can be used to reproduce the figures of this paper will be included in ESMValTool v2 at the time of publication of the paper.

#### **8** Author contributions

BG led the writing and analysis of the paper. MB and MR provided the satellite dataset. VE, PC and PF contributed to the evaluation of the CMIP simulations. All authors contributed to the writing of the manuscript.

#### 9 Competing interests

The authors declare that they have no conflict of interest.

#### **10** Acknowledgements

This work has been supported by ESA Climate Change Initiative (CCI) Climate Modelling User Group (CMUG) project, the

- Horizon 2020 project Climate-Carbon Interactions in the Current Century (4C) funded by the EU (Grant agreement ID 821003) and the EVal4CMIP project funded by the Helmholtz Society. The generation of the satellite XCO<sub>2</sub> data product has received funding from ESA (GHG-CCI) and the EU (Copernicus Climate Change Service C3S, led by ECMWF). We acknowledge the World Climate Research Programme (WCRP), which, through its Working Group on Coupled Modelling, coordinated and promoted CMIP6. We thank the climate modeling groups (listed in Tables 2 and 3 of this paper) for producing and making
- available their model output, the Earth System Grid Federation (ESGF) for archiving the data and providing access, and the multiple funding agencies who support CMIP and ESGF.

We also thank both anonymous reviewers for their helpful comments on this paper and the handling editor Trevor Keenan for taking on this paper.

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

### Tables

| Code | Location       | Latitude [°] | Longitude [°] | Altitude [m] | Start Year |
|------|----------------|--------------|---------------|--------------|------------|
| ASK  | Assekrem,      | 23.2625      | 5.6322        | 2710         | 1995       |
|      | Algeria        |              |               |              |            |
| CGO  | Cape Grim,     | -40.6800     | 144.6800      | 94           | 1984       |
|      | Australia      |              |               |              |            |
| LEF  | Park Falls,    | 45-945       | 269.7300      | 868          | 1994       |
|      | United States  |              |               |              |            |
| HUN  | Hegyhatsal,    | 46.950       | 16.650        | 248          | 1993       |
|      | Hungary        |              |               |              |            |
| WIS  | Ketura, Israel | 30.8595      | 34.7809       | 482          | 1995       |

### Table 1: List of active NOAA surface flask measurement sites used in this study.

Table 2: CMIP5 models analysed in this study. Under Comments, D stands for models including dynamic vegetation, and N for models including Nitrogen cycles.

| Model          | Institute                                                                                                               | Atmosphere<br>Model       | Land<br>Model              | Ocean Model         | Com<br>ment | Main Reference                               |
|----------------|-------------------------------------------------------------------------------------------------------------------------|---------------------------|----------------------------|---------------------|-------------|----------------------------------------------|
| BNU-ESM        | College of Global<br>Change and Earth<br>System Science,                                                                | CAM3.5                    | CoLM +<br>BNU-<br>DGVM     | MOM4p1 +<br>IBGC    | N, D        | Ji et al. (2014)                             |
| CanESM2        | Canadian Center for<br>Climate Modeling and<br>Analysis, BC, Canada                                                     | CanAM4                    | CLASS2.7<br>+ CTEM1        | СМОС                |             | Arora et al. (2011)                          |
| CESM1-BGC      | National Center for<br>Atmospheric Research<br>Boulder, CO, USA                                                         | CAM4                      | CLM4                       | POP2 + BEC          | N           | Gent et al. (2011); Lindsay<br>et al. (2014) |
| FIO-ESM        | The First Institute of<br>Oceanography, SOA,<br>China                                                                   | CAM3.0                    | CLM3.5 +<br>CASA           | POP2.0 +<br>OCMIP-2 |             | Bao et al. (2012); Qiao et<br>al. (2013)     |
| GFDL-ESM2G     | Geophysical Fluid<br>Dynamics Laboratory,<br>United States                                                              | AM2                       | LM3.0                      | GOLD +<br>TOPAZ2    | D           | Dunne et al. (2012); Dunne<br>et al. (2013)  |
| GFDL-<br>ESM2M | Geophysical Fluid<br>Dynamics Laboratory,<br>United States                                                              | AM2                       | LM3.0                      | MOM4.1 +<br>TOPAZ2  | D           | Dunne et al. (2012); Dunne et al. (2013)     |
| MIROC-ESM      | Japan Agency for<br>Marine-Earth Science<br>and Technology, Japan;<br>Atmosphere and Ocean<br>Research Institute, Japan | MIROC-AGCM<br>+ SPRINTARS | MATSIRO<br>+ SEIB-<br>DGVM | COCO3.4 +<br>NPZD   | D           | Watanabe et al. (2011)                       |
| MPI-ESM-LR     | Max Planck Institute for<br>Meteorology, Hamburg,<br>Germany                                                            | ECHAM6                    | JSBACH +<br>BETHY          | MPIOM +<br>HAMOCC5  | D           | Giorgetta et al. (2013)                      |

| MRI-ESM1   | Meteorological Research<br>Institute, Japan | MRI-AGCM3.3<br>+ MASINGAR<br>mk-2 + MRI-<br>CCM2 | HAL  | MRI.COM3 | D | Adachi et al. (2013);<br>Yukimoto et al. (2012);<br>Yukimoto et al. (2011) |
|------------|---------------------------------------------|--------------------------------------------------|------|----------|---|----------------------------------------------------------------------------|
| NorESM1-ME | Norwegian Climate<br>Center, Norway         | CAM4-Oslo                                        | CLM4 | HAMOCC5  | N | Tjiputra et al. (2013)                                                     |

Table 3: CMIP6 models analysed in this study. Under Comments, D stands for models including dynamic vegetation, and N for models including Nitrogen cycles.

| Model             | Institute                                                                        | Atmosphere<br>Model                                       | Land<br>Model                 | Ocean Model                         | Com<br>ment | Main Reference<br>& Data DOI                                           |
|-------------------|----------------------------------------------------------------------------------|-----------------------------------------------------------|-------------------------------|-------------------------------------|-------------|------------------------------------------------------------------------|
| ACCESS-<br>ESM1-5 | Commonwealth<br>Scientific and Industrial<br>Research Organisation,<br>Australia | UM7.3                                                     | CABLE2.4<br>with CASA-<br>CNP | MOM5 +<br>WOMBAT                    | N           | Law et al. (2017); Ziehn et<br>al. (2017)<br>Data: Ziehn et al. (2019) |
| CanESM5           | Canadian Center for<br>Climate Modeling and<br>Analysis, BC, Canada              | CanAM5                                                    | CLASS-<br>CTEM                | NEMO 3.4.1.<br>+ CMOC               |             | Swart et al. (2019a)<br>Data: Swart et al. (2019c)                     |
| CanESM5-<br>CanOE | Canadian Center for<br>Climate Modeling and<br>Analysis, BC, Canada              | CanAM5                                                    | CLASS-<br>CTEM                | NEMO 3.4.1.<br>+ CanOE              |             | Swart et al. (2019a)<br>Data: Swart et al. (2019b)                     |
| CNRM-ESM2-<br>1   | CNRM-CERFACS,<br>France                                                          | ARPEGE-<br>Climat v6.3 +<br>SURFEX v8.0                   | ISBA +<br>CTRIP               | NEMO v3.6 +<br>GELATO +<br>PISCESv2 |             | Séférian et al. (2019)<br>Data: Seferian (2019)                        |
| GFDL-ESM4         | Geophysical Fluid<br>Dynamics Laboratory,<br>United States                       | AM4.1                                                     | LM4.1                         | OM4 MOM6<br>+ COBALTv2              | D           | Dunne et al. (2020)<br>Data: Krasting et al.<br>(2018)                 |
| MIROC-ES2L        | MIROC, Japan                                                                     | MIROC-AGCM<br>+ SPRINTARS                                 | VISIT-e &<br>MATSIRO6         | COCO +<br>OECO v2                   | N           | Hajima et al. (2020b)<br>Data: Hajima et al.<br>(2020a)                |
| MPI-ESM1-2-<br>LR | Max Planck Institute for<br>Meteorology, Hamburg,<br>Germany                     | ECHAM6.3                                                  | JSBACH3.2                     | MPIOM1.6 +<br>HAMOCC6               | N, D        | Mauritsen et al. (2019)<br>Data: Wieners et al.<br>(2019)              |
| MRI-ESM2-0        | Meteorological Research<br>Institute, Japan                                      | MRI-<br>AGCM3.5 +<br>MASINGAR<br>mk-2r4c + MRI-<br>CCM2.1 | HAL                           | MRI.COMv4                           |             | Yukimoto et al. (2019a)<br>Data: Yukimoto et al.<br>(2019b)            |

| NorESM2-LM      | Norwegian<br>Center, Norv | Climate                 | Modified<br>CAM6 | CLM5             | HAMOCC             | N    | Seland et al. (2020)<br>Data: Seland et al. (2019) |
|-----------------|---------------------------|-------------------------|------------------|------------------|--------------------|------|----------------------------------------------------|
| UKESM1-0-<br>LL | Met Offic<br>Centre, Unit | ee Hadley<br>ed Kingdom | Unified Model    | JULES-ES-<br>1.0 | NEMO +<br>MEDUSA-2 | N, D | Sellar et al. (2019)<br>Data: Tang et al. (2019)   |

## Figures