# Peer review of "Spatially resolved evaluation of Earth system models with satellite column averaged CO2"

_Biogeosciences, 2020_

## Referee Comment (RC1) · Anonymous Referee #1 · 14 Jul 2020

This is a concise analysis of the current CMIP6-generation emission-driven Earth System Models' ability to reproduce satellite-observed variability characteristics of column-average atmospheric carbon dioxide concentrations (XCO2). The manuscript provides a comparison with the previous generation of models and demonstrates improvement over time as a modeling community. The manuscript also demonstrates that geographic and temporal sampling biases in the satellite observations contribute to an observed negative trend in the amplitude of the seasonal cycle of XCO2. This manuscript is an important documentation of the models' ability to simulate atmospheric CO2 and could be suitable for publication after addressing some of the concerns outlined below.

Major Comments:

1. The spatial sampling issues comparing models to satellite obs are addressed in
this manuscript, but the temporal issues are only partially addressed. It is not clear what role the presence of cloud cover plays in the results. Satellite observations of column-average CO2 occur over locations with low cloud cover (line 120). One could imagine that some processes – such as stomatal conductance – could vary significantly on cloudy vs. cloud-free days. The model monthly averages however include all model timesteps and are not impacted by the presence of clouds. Some quantitative assessment of this effect is needed to interpret the results of this study. Perhaps reconstructing some monthly averages using a daily or sub-daily cloud mask could help understand whether or not this has a large influence on the comparison between models and satellite observations.

2. What are the baseline trends in the control simulations for the physical climate and carbon cycle processes that influence atmospheric CO2? Was there an attempt to detrend the models? Why or why not?

3. Were multiple model ensemble members from each model considered? The manuscript seems to suggest that only one ensemble member from each model was considered. This point should be clarified and all available ensemble members should be analyzed to get as comprehensive a picture as possible regarding the models' intrinsic variability given the nature of this study and the relatively short time period of analysis.

4. The conclusions section would benefit from a longer discussion regarding the limitations of the study and future directions. Can the authors make any further recommendations regarding improvements that are needed on either the observational or modeling side to make this comparison better?

Minor Comments: * Line 21: Replace "slightly" with a more quantitative value * Line 40: Unequivocal warming of what? Troposphere? * Lines 46-48: This sentence has some grammatical issues * Line 57: What is meant by "seems to be" * Line 119: The observational record is already relatively short from a climate perspective. Discarding

2 years seems like a lot. Consider adding in the years if simulations are now available
* Lines 131-133: Consider expanding the discussions as to why these sites were se-
lected * Line 205-206: Consider mentioning these offsets sooner in the paragraph to
improve readability.

References: * Please add data DOIs for all CMIP6 datasets downloaded and analyzed
from the ESGF archive.

Figure Comments: * Figure 3: Is there a way to incorporate linear trend information
into this figure? * Figures 6a & 6b: Consistent color scale ranges are needed for
comparison * Figures 7a and 8a: A non-diverging color scale for the top panels could
make it easier to contrast against the information contained in the bottom panels

---

## Referee Comment (RC2) · Anonymous Referee #2 · 16 Jul 2020

This study used satellite-observed column-average CO$_2$ (XCO$_2$) to evaluate how well the current generation of earth system models reproduces atmospheric CO$_2$ variability. The authors compared spatially resolved model simulations of XCO$_2$ with observed XCO$_2$ in terms of biases, growth rates, seasonal cycle amplitudes (SCA), and trends in SCA. They found that most models overestimate XCO$_2$ and the growth rates of XCO$_2$, but underestimate the seasonal amplitudes of XCO$_2$. The study is novel and interesting, and could be considered for publication after concerns are addressed.

[Figure]

**General comments**

1. The apparent "trend reversal" of SCA in satellite $XCO_2$ caused by the sampling coverage bias raises an important question whether other characteristics of the observed $XCO_2$ were subject to the same bias. It seems that the "sampled" vs. "unsampled" comparison was done only for the trend in SCA. Would this comparison give different results for the growth rate of $XCO_2$?

2. Although the effect of different spatial coverage between the two instruments has been addressed in Figs. 9 and 10, there has been no mention of whether the measurement scales of $XCO_2$ were calibrated between the two instruments. This could be another source of bias. During the overlapping period, did SCIAMACHY and GOSAT measurements agree well with each other?

3. In addition to the spatial distribution of SCA shown in Fig. 6, it would also be interesting to compare the simulated latitudinal gradients (i.e., the zonal mean) of SCA from different models on the same plot.

4. Given that there was no correlation between the growth rate and the growing-season temperature anomaly in the *observations* (Fig. 5), I'm not convinced that a robust emergent constraint relationship can be established for the period of 2003–2014. Note that this is different from Fig. 2 in Cox et al. (2013) *Nature* in which a clear correlation is seen in the observations. In the absence of a relationship of the same kind in the observations, one could not distinguish a real emergent constraint from an artifact of model assumptions. It may as well exist for a longer period with more data, but we couldn't tell. I suggest moving the paragraph of P8L248–P9L263 as well as Fig. 5 to the supplement.

5. In all figures that show a regression line, it would be better to show both the Pearson correlation and the $p$-value.

6. The study would benefit from a more structured discussion of lessons learned from model evaluation, for example, what the likely causes of model biases in $XCO_2$ and the SCA of $XCO_2$ would be. There are some good points made in section 5.3 and the summary, which could be better organized.

**Specific comments**

- Abstract: The abstract is quite long and technical. Describe the major findings concisely and leave nonessential details to the main text.

- P1L20: The multi-model means give a feeling that CMIP6 models had a great improvement relative to CMIP5 models. But a closer look at Fig. 3 would tell that this was mainly because negative biases canceled positive biases. I suggest adding the multi-model standard deviations of the bias (or other statistics that characterize the spread) in parentheses.

- P1L25–32: The "trend reversal" in SCA caused by sampling bias is not clear on a first reading. It would be helpful to write this in a way that is less entangled.

- P2L46–60: If the purpose of this paragraph is to introduce SCA, then the first few sentences seem quite redundant. Better get to the point straight away.

- P2L64: Missing Ciais et al., 2013 in the References.

- P3L65: "downlooking" → "downward-looking"

- P3L89–90: "such as a general overestimation of photosynthesis [relative to data-driven models]." We don't know the magnitude of the global photosynthesis with certainty. The number could range from 112 to 169 PgC yr$^{-1}$ (Ryu et al., 2019, *RSE*). The MTE GPP data product (Jung et al., 2011, *JGR*) that was used to evaluate the CMIP models would sit near the lower end of this range.
- P5L131: Missing Dlugokencky et al., 2018 in the References.

- P7L203: "parameters" → "variables"

- P7L205: Surface observations sample the air within or closer to the boundary layer, and therefore may have a larger seasonal swing.

- P8L241: What is the correlation between the observed GR and the multi-model mean GR?

- P9L256: The increase of GR sensitivity to temperature after including 2015 and 2016 data could have been due to El Niño.

- Table 1: Use whole numbers in the "Altitude" column.

- Figure 2: I suggest lightening the gray background to prevent it from interfering with the reading of the curves.

- Figure 3: It seems that after detrending, the seasonal variability in the multi-model mean $XCO_2$ would match quite well with that in the observed $XCO_2$, and better than the seasonal variability in any individual models. I wonder what the correlation coefficients would be.

- Figure 4: I think ranking the models by their average growth rates, from low to high, would make the figure clearer.

- Figures 6a and 6b: Why not use the same scale? One could reserve the purple region of the colormap for the high SCA values from MPI-ESM-LR. This would not affect other models that have values represented in colors from blue to red.

---

## Author Comment (AC2) · 4 Sep 2020

**Response to Anonymous Referee #2**

We thank the reviewer for the helpful comments. We have revised the manuscript according to all review comments we have received. A pointwise reply is given below, with the original comments in **bold**, and our answers in red.

**This study used satellite-observed column-average CO$_2$ (XCO$_2$) to evaluate how well the current generation of earth system models reproduces atmospheric CO$_2$ variability. The authors compared spatially resolved model simulations of XCO2 with observed XCO$_2$ in terms of biases, growth rates,**

seasonal cycle amplitudes (SCA), and trends in SCA. They found that most models overestimate $XCO_2$ and the growth rates of $XCO_2$, but underestimate the seasonal amplitudes of $XCO_2$. The study is novel and interesting, and could be considered for publication after concerns are addressed.

We thank Referee #2 for the constructive comments which helped to improve the manuscript.

**General comments**
1. The apparent "trend reversal" of SCA in satellite $XCO_2$ caused by the sampling coverage bias raises an important question whether other characteristics of the observed $XCO_2$ were subject to the same bias. It seems that the "sampled" vs. "unsampled" comparison was done only for the trend in SCA. Would this comparison give different results for the growth rate of XCO2?

As mentioned in section 5.2. the spatial variability of the growth rate is small. We did analyze the sampling on the other considered quantities, but found no significant sampling impact on the growth rate. We have also added a panel to Figure 3 which depicts the mean monthly growth rate, and while the change from the satellites is visible in these monthly rates in 2009, this is due to the way in which it is calculated and averaged out in the annual values. The annual growth rates of the satellite dataset have been found to correlate well (correlation coefficient of 0.82) with the NOAA global growth rates (Buchwitz et al., 2018). We have added the sentence "No significant changes to the annual growth rates due to the satellite spatial coverage have been found." in section 5.2 to clarify this.

2. Although the effect of different spatial coverage between the two instruments has been addressed in Figs. 9 and 10, there has been no mention of whether the measurement scales of $XCO_2$ were calibrated between the two

**instruments. This could be another source of bias. During the overlapping period, did SCIAMACHY and GOSAT measurements agree well with each other?**
The joint time series is computed using the Ensemble Median Algorithm (EMMA, Reuter et al. (2013); Reuter et al. (2020)), which includes a bias correction to all products during overlap phases, resulting in a good agreement during the overlap period. This bias correction is shown in Figure 5 of Reuter et al. (2020).

**3. In addition to the spatial distribution of SCA shown in Fig. 6, it would also be interesting to compare the simulated latitudinal gradients (i.e., the zonal mean) of SCA from different models on the same plot.**

We have added zonal mean panels to the map plots in Figure 6. They show the increase of SCA with increasing latitude for all models, with growth spurts around the equator and in the midlatitudes for most models.

**4. Given that there was no correlation between the growth rate and the growing-season temperature anomaly in the observations (Fig. 5), I'm not convinced that a robust emergent constraint relationship can be established for the period of 2003–2014. Note that this is different from Fig. 2 in Cox et al. (2013) Nature in which a clear correlation is seen in the observations. In the absence of a relationship of the same kind in the observations, one could not distinguish a real emergent constraint from an artifact of model assumptions. It may as well exist for a longer period with more data, but we couldn't tell. I suggest moving the paragraph of P8L248–P9L263 as well as Fig. 5 to the supplement.**

We agree with this assessment, as discussed in ll. 257-259 - "This shows that the time period 2003–2014 is not sufficient to reproduce the emergent constraint, but it might be feasible once CMIP6 emission driven future simulations are available for a longer time overlap between models and observations". We have moved this

paragraph and its accompanying Figure to Appendix C.

**5.   In all figures that show a regression line, it would be better to show both the Pearson correlation and the p-value.**

p-values have been added next to the Pearson correlation coefficient in the bottom right of the panels.

**6.   The study would benefit from a more structured discussion of lessons learned from model evaluation, for example, what the likely causes of model biases in $XCO_2$ and the SCA of $XCO_2$ would be. There are some good points made in section 5.3 and the summary, which could be better organized.**

Following a suggestion from Referee #1 we have added a paragraph devoted to the discussion on the limitations and future directions, which should include some of these points.
"There are several ways to improve on this analysis in the future. With more available future scenario simulations, the analysis can be extended for a longer time series, making use of longer observational timeseries, such as the one introduced in Reuter et al. (2020). Higher temporal resolution of the models would enable studies on the effect of the diurnal cycle of $CO_2$ on the monthly mean and also allow for the construction of a co-located time series with the Level 2 satellite data. This could help highlight some of the causes of model biases by being able to pinpoint time and space where they occur more precisely. Model biases may also result from the CMIP experimental design, such as requiring the climate state to be in equilibrium in 1850 while the real world may not have been (Bronselaer et al., 2017), or the parameterizations of biological and physical processes not allowing the system to change rapidly enough (Hoffman et al., 2014). Along with a longer time series, newer satellites, such as OCO-2 or the planned Sentinel 7 bring higher resolutions and more data, potentially helping

to fill in the gaps and reduce the impact of the sampling we discussed in Section 5.3.2.".

**Specific comments**
**• Abstract: The abstract is quite long and technical. Describe the major findings concisely and leave nonessential details to the main text.**
We have shortened the abstract by removing several of the technical sentences which are not required to understand the results.

**• P1L20: The multi-model means give a feeling that CMIP6 models had a great improvement relative to CMIP5 models. But a closer look at Fig. 3 would tell that this was mainly because negative biases canceled positive biases. I suggest adding the multi-model standard deviations of the bias (or other statistics that characterize the spread) in parentheses.**
We have added the full model spread alongside the means.

**• P1L25–32: The "trend reversal" in SCA caused by sampling bias is not clear on a first reading. It would be helpful to write this in a way that is less entangled.**
We have rewritten this part as follows to make it clearer: "While the combined satellite product shows a strong negative trend of decreasing effective SCA with increasing $XCO_2$ in the northern midlatitudes, both CMIP ensembles instead show a non-significant positive trend in the multi-model mean. The negative trend is reproduced by the models when sampling them as the observations, attributing it to sampling characteristics. Applying a mask of the mean data coverage of each satellite to the models, the effective SCA is higher for the SCIAMACHY/ENVISAT mask than when using the TANSO-FTS/GOSAT mask. This induces an artificial negative trend when using observational sampling over the full period, as SCIAMACHY/ENVISAT covers the early period until 2012, with TANSO-FTS/GOSAT measurements starting in 2009."

**• P2L46–60: If the purpose of this paragraph is to introduce SCA, then the first few sentences seem quite redundant. Better get to the point straight away.**

We have rewritten this part to be a more concise introduction of SCA: "Photosynthesis causes a net uptake of atmospheric $CO_2$ and thus declining atmospheric $CO_2$ concentrations in the growing season. Conversely, atmospheric $CO_2$ concentrations rise throughout the dormant season when there is a net release of $CO_2$ from the land due to decomposition of organic matter in soils. This uptake and release of carbon by the terrestrial biosphere throughout the year causes a seasonal cycle of atmospheric $CO_2$ (Keeling et al., 1989)."

**• P2L64: Missing Ciais et al., 2013 in the References.**

Fixed issue which resulted in reference not showing in the bibliography, thank you for spotting this.

**• P3L65: "downlooking" ⟶ "downward-looking"**

Changed.

**• P3L89–90: "such as a general overestimation of photosynthesis [relative to datadriven models]." We don't know the magnitude of the global photosynthesis with certainty. The number could range from 112 to 169 PgC $yr^{-1}$ (Ryu et al., 2019, RSE). The MTE GPP data product (Jung et al., 2011, JGR) that was used to evaluate the CMIP models would sit near the lower end of this range.**

We have added this as a caveat and added the example of the overestimated leaf area index. "... such as a general overestimation leaf area index and photosynthesis. However, the magnitude of the global photosynthesis is not well constrained by observations, with estimates ranging between 112 and 169 PgC $yr^{-1}$ (Ryu et al., 2019), with the dataset used by Anav et al. (2013) on the lower end of this range."

**P5L131: Missing Dlugokencky et al., 2018 in the References.**
Updated this reference to newer version and solved issue preventing it from showing in bibliography, thank you for spotting this.

**• P7L203: "parameters" ⟶ "variables"**
Changed.

**• P7L205: Surface observations sample the air within or closer to the boundary layer, and therefore may have a larger seasonal swing.**
The larger seasonal swing and thus higher seasonal cycle amplitude at the surface is discussed in the next paragraph. L205 was referring to the total offset of the mean, as evident most clearly in the Cape Grim and Hegyhatsal stations, comparing the blue (station) and red (model surface) curves.

**• P8L241: What is the correlation between the observed GR and the multi-model mean GR?**
The correlation between the observed and multi-model mean annual GR is 0.48 in CMIP6 and 0.07 in CMIP5, which is higher than most the models. We have added this to the paragraph.

**• P9L256: The increase of GR sensitivity to temperature after including 2015 and 2016 data could have been due to El Niño.**
This is true and has been added as a note in this sentence (now in the Appendix), as well as mentioned as a reason for suggesting that the timeseries is not long enough to support trying to reproduce this emergent constraint.

**• Table 1: Use whole numbers in the "Altitude" column.**
Changed to whole numbers.

**• Figure 2: I suggest lightening the gray background to prevent it from interfering with the reading of the curves.**
Changed the color of the land areas of the underlaying map plot to a very light grey.

**• Figure 3: It seems that after detrending, the seasonal variability in the multimodel mean XCO$_2$ would match quite well with that in the observed XCO$_2$, and better than the seasonal variability in any individual models. I wonder what the correlation coefficients would be.**
We have extended this figure with panels showing the calculated monthly growth rate and the detrended seasonal cycle. The correlation between the detrended seasonal cycle of the multi-model mean to the observations is 0.98 in CMIP6 and 0.93 in CMIP5. in CMIP6 this is the best correlation, the closest models have a correlation of 0.96 (MRI-ESM2-0 and GFDL-ESM4). For CMIP6 a few models are close and a bit higher than the multi-model mean with 0.93 for NorESM1-ME and MIROC-ESM, 0.94 for MRI-ESM1 and 0.95 for MPI-ESM-LR. So the multi-model mean in CMIP6 indeed captures the seasonal variability better than any individual model.

**• Figure 4: I think ranking the models by their average growth rates, from low to high, would make the figure clearer.**
Models are now ranked from low to high in the barplot.

**• Figures 6a and 6b: Why not use the same scale? One could reserve the purple region of the colormap for the high SCA values from MPI-ESM-LR. This would not affect other models that have values represented in colors from blue to red.**
We have adjusted the scale to be the same for both figures. We have furthermore changed the colormap to a non-diverging one, which was requested by Referee #1 for the top panels in figure 6 (formerly 7) to contrast the bottom panels showing the

differences. This colormap has been adapted for Figure 5 (formerly 6) as well for consistency.

**References**

Anav, A., Friedlingstein, P., Kidston, M., Bopp, L., Ciais, P., Cox, P., Jones, C., Jung, M., Myneni, R., and Zhu, Z.: Evaluating the Land and Ocean Components of the Global Carbon Cycle in the CMIP5 Earth System Models, J Climate, 26, 6801-6843, 10.1175/Jcli-D-12-00417.1, 2013.

Bronselaer, B., Winton, M., Russell, J., Sabine, C. L., and Khatiwala, S.: Agreement of CMIP5 Simulated and Observed Ocean Anthropogenic CO2 Uptake, Geophys Res Lett, 44, 12,298-212,305, 10.1002/2017gl074435, 2017.

Buchwitz, M., Reuter, M., Schneising, O., Noel, S., Gier, B., Bovensmann, H., Burrows, J. P., Boesch, H., Anand, J., Parker, R. J., Somkuti, P., Detmers, R. G., Hasekamp, O. P., Aben, I., Butz, A., Kuze, A., Suto, H., Yoshida, Y., Crisp, D., and O'Dell, C.: Computation and analysis of atmospheric carbon dioxide annual mean growth rates from satellite observations during 2003-2016, Atmos Chem Phys, 18, 1-22, 10.5194/acp-18-17355-2018, 2018.

Hoffman, F. M., Randerson, J. T., Arora, V. K., Bao, Q., Cadule, P., Ji, D., Jones, C. D., Kawamiya, M., Khatiwala, S., Lindsay, K., Obata, A., Shevliakova, E., Six, K. D., Tjiputra, J. F., Volodin, E. M., and Wu, T.: Causes and implications of persistent atmospheric carbon dioxide biases in Earth System Models, J Geophys Res-Biogeo, 119, 141-162, 10.1002/2013jg002381, 2014.

Keeling, C. D., Bacastow, R. B., Carter, A. F., Piper, S. C., Whorf, T. P., Heimann, M., Mook, W. G., and Roeloffzen, H.: A three-dimensional model of atmospheric CO2 transport based on observed winds: 1. Analysis of observational data, in:

Aspects of Climate Variability in the Pacific and the Western Americas, Geophysical Monograph Series, 165-236, 1989.

Reuter, M., Bösch, H., Bovensmann, H., Bril, A., Buchwitz, M., Butz, A., Burrows, J. P., amp, apos, Dell, C. W., Guerlet, S., Hasekamp, O., Heymann, J., Kikuchi, N., Oshchepkov, S., Parker, R., Pfeifer, S., Schneising, O., Yokota, T., and Yoshida, Y.: A joint effort to deliver satellite retrieved atmospheric CO2 concentrations for surface flux inversions: the ensemble median algorithm EMMA, Atmos Chem Phys, 13, 1771-1780, 10.5194/acp-13-1771-2013, 2013.

Reuter, M., Buchwitz, M., Schneising, O., Noël, S., Bovensmann, H., Burrows, J. P., Boesch, H., Di Noia, A., Anand, J., Parker, R. J., Somkuti, P., Wu, L., Hasekamp, O. P., Aben, I., Kuze, A., Suto, H., Shiomi, K., Yoshida, Y., Morino, I., Crisp, D., O'Dell, C. W., Notholt, J., Petri, C., Warneke, T., Velazco, V. A., Deutscher, N. M., Griffith, D. W. T., Kivi, R., Pollard, D. F., Hase, F., Sussmann, R., Té, Y. V., Strong, K., Roche, S., Sha, M. K., De Mazière, M., Feist, D. G., Iraci, L. T., Roehl, C. M., Retscher, C., and Schepers, D.: Ensemble-based satellite-derived carbon dioxide and methane column-averaged dry-air mole fraction data sets (2003–2018) for carbon and climate applications, Atmos. Meas. Tech., 13, 789-819, 10.5194/amt-13-789-2020, 2020.

Ryu, Y., Berry, J. A., and Baldocchi, D. D.: What is global photosynthesis? History, uncertainties and opportunities, Remote Sens Environ, 223, 95-114, https://doi.org/10.1016/j.rse.2019.01.016, 2019.

---

## Author Response (AR1)

**Author's response – Gier et al. bg-2020-170**

**Response to Anonymous Referee #1**

We thank the reviewer for the helpful comments. We have revised the manuscript according to all review comments we have received. A pointwise reply is given below, with the original comments in bold black, and our answers in red.

5 **This is a concise analysis of the current CMIP6-generation emission-driven Earth System Models' ability to reproduce satellite-observed variability characteristics of column-average atmospheric carbon dioxide concentrations (XCO2). The manuscript provides a comparison with the previous generation of models and demonstrates improvement over time as a modeling community. The manuscript also demonstrates that geographic and temporal sampling biases in the satellite observations contribute to an observed negative trend in the amplitude of the seasonal**

10 **cycle of XCO2. This manuscript is an important documentation of the models' ability to simulate atmospheric CO2 and could be suitable for publication after addressing some of the concerns outlined below.**

We thank Referee #1 for the constructive comments which helped to improve the manuscript.

15 **Major Comments:**
**1. The spatial sampling issues comparing models to satellite obs are addressed in this manuscript, but the temporal issues are only partially addressed. It is not clear what role the presence of cloud cover plays in the results. Satellite observations of column-average CO2 occur over locations with low cloud cover (line 120). One could imagine that some processes – such as stomatal conductance – could vary significantly on cloudy vs. cloud-free days. The model**

20 **monthly averages however include all model timesteps and are not impacted by the presence of clouds. Some quantitative assessment of this effect is needed to interpret the results of this study. Perhaps reconstructing some monthly averages using a daily or sub-daily cloud mask could help understand whether or not this has a large influence on the comparison between models and satellite observations.**

25 Only monthly frequency $CO_2$ data is currently available on the ESGF and therefore an analysis as proposed here is not possible at the current time. While it is true that studies have found cloud cover to have an impact on photosynthesis, the response can be fundamentally different for various ecosystems (Still et al., 2009). Cheng et al. (2016) found that "the diffuse light effect from clouds is not as strong of a driver of regional or global ecosystem productivity in temperate ecosystems during the midday as previously suggested in other studies". The satellite data we use is measured at 13:00 local

30 time, which falls into this midday period. Moreover, we would expect a larger effect from the diurnal cycle than the cloud cover with the satellite data measure at 13:00, while the model monthly means are computed using both day and night data. While both of these effects may change the absolute values, their scale should not vary much throughout the years, so that they have no effect on relative changes on growth rate and seasonal cycle amplitude. We have added this point as a caveat in section 5.3.2. and in the Conclusions as part of the discussion of limitations and

35 future directions.

**2. What are the baseline trends in the control simulations for the physical climate and carbon cycle processes that influence atmospheric CO2? Was there an attempt to detrend the models? Why or why not?**

40 We have looked at various variables for a few sample models in the control simulations, and found no significant trend in the physical processes. For $CO_2$ the trend in the control simulations is negligible compared to the interannual variability, as discussed e.g. by Dunne et al. (2020) for GFDL-ESM4. Therefore, the models have not been detrended. A detailed analysis including all the processes which influence atmospheric $CO_2$ which you are suggesting here on a per model basis is a study on its own and beyond the scope of this paper.

**3. Were multiple model ensemble members from each model considered? The manuscript seems to suggest that only one ensemble member from each model was considered. This point should be clarified and all available ensemble members should be analyzed to get as comprehensive a picture as possible regarding the models' intrinsic variability given the nature of this study and the relatively short time period of analysis.**

In CMIP5 only one model performed the future scenario simulations with more than one ensemble member, and therefore we have chosen not to include these. For CMIP6 there are various models with several ensemble members. We have extended Figure 3 (the timeseries) both with additional panels depicting the computed monthly growth rate and detrended seasonal cycle, as well as including all the ensemble members for CMIP6 in it. The multi-model mean shown in this plot only includes the first member for each model. A deeper analysis shows that while there are small differences in the growth rate for different ensembles members, the SCA and its patterns on the map plots are very similar. The inclusion of more ensemble members does not impact the existing analysis and we have therefore elected to only include the first ensemble member for each model in all analysis beyond Figure 3, which gives a good overview of the models intrinsic variability. Using an ensemble mean would average out much of the interannual variability found in each individual member. We have made this clearer by ending section 2.2 on the model simulations with "For CMIP5, only one model had more than one ensemble member performing the emission driven RCP 8.5 simulation and thus only one ensemble member for each model has been used. In CMIP6, several models have three or more ensemble members. We consider all of them in Figure 3 for the timeseries to show the models' intrinsic variability, but then proceed the analysis with only the first ensemble member for each model, as they perform similarly to each other for the analysis in this paper, and using an ensemble mean would reduce the interannual variability found in each individual member."

**4. The conclusions section would benefit from a longer discussion regarding the limitations of the study and future directions. Can the authors make any further recommendations regarding improvements that are needed on either the observational or modeling side to make this comparison better?**

70

We have added a paragraph devoted to the discussion on the limitations and future directions:
"There are several ways to improve on this analysis in the future. With more available future scenario simulations, the analysis can be extended for a longer time series, making use of longer observational timeseries, such as the one introduced in Reuter et al. (2020). Higher temporal resolution of the models would enable studies on the effect of the diurnal cycle of $CO_2$ on the monthly mean and also allow for the construction of a co-located time series with the Level 2 satellite data. This could help highlight some of the causes of model biases by being able to pinpoint time and space where they occur more precisely. Model biases may also result from the CMIP experimental design, such as requiring the climate state to be in equilibrium in 1850 while the real world may not have been (Bronselaer et al., 2017), or the parametrizations of biological and physical processes not allowing the system to change rapidly enough (Hoffman et al., 2014). Along with a longer time series, newer satellites, such as OCO-2 or the planned Sentinel 7 bring higher resolutions and more data, potentially helping to fill in the gaps and reduce the impact of the sampling we discussed in Section 5.3.2."

**Minor Comments:**
**Line 21: Replace "slightly" with a more quantitative value**
85    Replaced "slightly" with the multi-model mean bias for the growth rate.

**Line 40: Unequivocal warming of what? Troposphere?**
Changed to "unequivocal warming of the climate system" which was used in the IPCC report used as reference.

90    **Lines 46-48: This sentence has some grammatical issues**
Following a suggestion by referee #2 to shorten this part of the introduction, this sentence has been removed.

**Line 57: What is meant by "seems to be"**

Changed sentence to "Although models do not agree unanimously, the dominant effects are a positive trend in SCA due to the $CO_2$ fertilization combined with a negative trend due to climate warming."

95

**Line 119: The observational record is already relatively short from a climate perspective. Discarding 2 years seems like a lot. Consider adding in the years if simulations are now available.**
While we have been able to add additional models to the analysis, scenario simulations are still not available for all models discussed in this study and have therefore not been included.

100

**Lines 131-133: Consider expanding the discussions as to why these sites were selected**
The discussion has been reworded to make the selection process clearer, with stronger critera being mentioned first. "Measurement sites at locations with no available satellite data were excluded from the analysis, which ruled out the four baseline observatories in Mauna Loa, Samoa, as well as the South Pole and Point Barrow sites. Furthermore, sites which did not collect data during the period from 2003–2014 were discarded. From the remaining sites, a sample of five sites was chosen which had the best coverage of different latitudes, and when latitudes were similar, different longitudes were selected for increased spatial coverage. The selected sites are listed in Table 1."

105

110 **Line 205-206: Consider mentioning these offsets sooner in the paragraph to improve readability.**
Swapped this and the previous sentence.

**References: Please add data DOIs for all CMIP6 datasets downloaded and analyzed from the ESGF archive.**
115 Data citations with DOIs have been added as the last entry in Table 2 under "References".

**Figure Comments:**
**Figure 3: Is there a way to incorporate linear trend information into this figure?**
As mentioned above, we have added additional panels to this figure, showing the growth rate and the detrended seasonal
120 cycle. As the growth rate symbolizes the trend and the mean value with interannual variability is given in Figure 4, we believe this is enough. Adding regression lines and further linear trend information to the time series panel would clutter the figure.

**Figures 6a & 6b: Consistent color scale ranges are needed for comparison**
125 Implemented consistent color scale for both CMIP ensembles. We have also changed the color scale to the non-divergent newly implemented one used for the top panels of Figure 6 (formerly 7) for consistency.

**Figures 7a and 8a: A non-diverging color scale for the top panels could make it easier to contrast against the information contained in the bottom panels**
130 Changed the color scale to a non-divergent one for the top panels in Figure 6 (formerly 7).

References
Bronselaer, B., Winton, M., Russell, J., Sabine, C. L., and Khatiwala, S.: Agreement of CMIP5 Simulated and Observed Ocean Anthropogenic CO2 Uptake, Geophys Res Lett, 44, 12,298-212,305, 10.1002/2017gl074435, 2017.

135 Cheng, S. J., Steiner, A. L., Hollinger, D. Y., Bohrer, G., and Nadelhoffer, K. J.: Using satellite-derived optical thickness to assess the influence of clouds on terrestrial carbon uptake, Journal of Geophysical Research: Biogeosciences, 121, 1747-1761, 10.1002/2016jg003365, 2016.

Dunne, J. P., Horowitz, L. W., Adcroft, A. J., Ginoux, P., Held, I. M., John, J. G., Krasting, J. P., Malyshev, S., Naik, V., Paulot, F., Shevliakova, E., Stock, C. A., Zadeh, N., Balaji, V., Blanton, C., Dunne, K. A., Dupuis, C., Durachta, J., Dussin,
140 R., Gauthier, P. P. G., Griffies, S. M., Guo, H., Hallberg, R. W., Harrison, M., He, J., Hurlin, W., McHugh, C., Menzel, R., Milly, P. C. D., Nikonov, S., Paynter, D. J., Ploshay, J., Radhakrishnan, A., Rand, K., Reichl, B. G., Robinson, T.,

Schwarzkopf, D. M., Sentman, L. T., Underwood, S., Vahlenkamp, H., Winton, M., Wittenberg, A. T., Wyman, B., Zeng, Y., and Zhao, M.: The GFDL Earth System Model version 4.1 (GFDL-ESM 4.1): Overall coupled model description and simulation characteristics, J Adv Model Earth Sy, n/a, e2019MS002015, 10.1029/2019ms002015, 2020.

145  Hoffman, F. M., Randerson, J. T., Arora, V. K., Bao, Q., Cadule, P., Ji, D., Jones, C. D., Kawamiya, M., Khatiwala, S., Lindsay, K., Obata, A., Shevliakova, E., Six, K. D., Tjiputra, J. F., Volodin, E. M., and Wu, T.: Causes and implications of persistent atmospheric carbon dioxide biases in Earth System Models, J Geophys Res-Biogeo, 119, 141-162, 10.1002/2013jg002381, 2014.

Reuter, M., Buchwitz, M., Schneising, O., Noël, S., Bovensmann, H., Burrows, J. P., Boesch, H., Di Noia, A., Anand, J., 150  Parker, R. J., Somkuti, P., Wu, L., Hasekamp, O. P., Aben, I., Kuze, A., Suto, H., Shiomi, K., Yoshida, Y., Morino, I., Crisp, D., O'Dell, C. W., Notholt, J., Petri, C., Warneke, T., Velazco, V. A., Deutscher, N. M., Griffith, D. W. T., Kivi, R., Pollard, D. F., Hase, F., Sussmann, R., Té, Y. V., Strong, K., Roche, S., Sha, M. K., De Mazière, M., Feist, D. G., Iraci, L. T., Roehl, C. M., Retscher, C., and Schepers, D.: Ensemble-based satellite-derived carbon dioxide and methane column-averaged dry-air mole fraction data sets (2003–2018) for carbon and climate applications, Atmos. Meas. Tech., 13, 789-819, 10.5194/amt-13-155  789-2020, 2020.

Still, C. J., Riley, W. J., Biraud, S. C., Noone, D. C., Buenning, N. H., Randerson, J. T., Torn, M. S., Welker, J., White, J. W. C., Vachon, R., Farquhar, G. D., and Berry, J. A.: Influence of clouds and diffuse radiation on ecosystem-atmosphere $CO_2$ and $CO_{18}O$ exchanges, Journal of Geophysical Research: Biogeosciences, 114, 10.1029/2007jg000675, 2009.

160

**Response to Anonymous Referee #2**

We thank the reviewer for the helpful comments. We have revised the manuscript according to all review comments we have received. A pointwise reply is given below, with the original comments in bold black, and our answers in red.

**This study used satellite-observed column-average CO2 (XCO2) to evaluate how well the current generation of earth system models reproduces atmospheric CO2 variability. The authors compared spatially resolved model simulations of XCO2 with observed XCO2 in terms of biases, growth rates, seasonal cycle amplitudes (SCA), and trends in SCA. They found that most models overestimate XCO2 and the growth rates of XCO2, but underestimate the seasonal amplitudes of XCO2. The study is novel and interesting, and could be considered for publication after concerns are addressed.**

We thank Referee #2 for the constructive comments which helped to improve the manuscript.

**General comments**
**1. The apparent "trend reversal" of SCA in satellite XCO2 caused by the sampling coverage bias raises an important question whether other characteristics of the observed XCO2 were subject to the same bias. It seems that the "sampled" vs. "unsampled" comparison was done only for the trend in SCA. Would this comparison give different results for the growth rate of XCO2?**

As mentioned in section 5.2. the spatial variability of the growth rate is small. We did analyze the sampling on the other considered quantities, but found no significant sampling impact on the growth rate. We have also added a panel to Figure 3 which depicts the mean monthly growth rate, and while the change from the satellites is visible in these monthly rates in 2009, this is due to the way in which it is calculated and averaged out in the annual values. The annual growth rates of the satellite dataset have been found to correlate well (correlation coefficient of 0.82) with the NOAA global growth rates (Buchwitz et al., 2018). We have added the sentence "No significant changes to the annual growth rates due to the satellite spatial coverage have been found." in section 5.2 to clarify this.

**2. Although the effect of different spatial coverage between the two instruments has been addressed in Figs. 9 and 10, there has been no mention of whether the measurement scales of XCO2 were calibrated between the two instruments. This could be another source of bias. During the overlapping period, did SCIAMACHY and GOSAT measurements agree well with each other?**

The joint time series is computed using the Ensemble Median Algorithm (EMMA, Reuter et al. (2013);Reuter et al. (2020)), which includes a bias correction to all products during overlap phases, resulting in a good agreement during the overlap period. This bias correction is shown in Figure 5 of (Reuter et al., 2020).

**3. In addition to the spatial distribution of SCA shown in Fig. 6, it would also be interesting to compare the simulated latitudinal gradients (i.e., the zonal mean) of SCA from different models on the same plot.**

We have added zonal mean panels to the map plots in Figure 6. They show the increase of SCA with increasing latitude for all models, with growth spurts around the equator and in the midlatitudes for most models.

**4. Given that there was no correlation between the growth rate and the growing-season temperature anomaly in the *observations* (Fig. 5), I'm not convinced that a robust emergent constraint relationship can be established for the period of 2003–2014. Note that this is different from Fig. 2 in Cox et al. (2013) Nature in which a clear correlation is seen in the observations. In the absence of a relationship of the same kind in the observations, one could not distinguish a real emergent constraint from an artifact of model assumptions. It may as well exist for a longer period with more data, but we couldn't tell. I suggest moving the paragraph of P8L248–P9L263 as well as Fig. 5 to the supplement.**

210 We agree with this assessment, as discussed in l. 257-259 - "This shows that the time period 2003–2014 is not sufficient to reproduce the emergent constraint, but it might be feasible once CMIP6 emission driven future simulations are available for a longer time overlap between models and observations". We have moved this paragraph and its accompanying Figure to Appendix C.

215 **5. In all figures that show a regression line, it would be better to show both the Pearson correlation and the p-value.**

p-values have been added next to the Pearson correlation coefficient in the bottom right of the panels.

**6. The study would benefit from a more structured discussion of lessons learned from model evaluation, for example,**
220 **what the likely causes of model biases in XCO2 and the SCA of XCO2 would be. There are some good points made in section 5.3 and the summary, which could be better organized.**

Following a suggestion from Referee #1 we have added a paragraph devoted to the discussion on the limitations and future directions, which should include some of these points.
225 "There are several ways to improve on this analysis in the future. With more available future scenario simulations, the analysis can be extended for a longer time series, making use of longer observational timeseries, such as the one introduced in Reuter et al. (2020). Higher temporal resolution of the models would enable studies on the effect of the diurnal cycle of $CO_2$ on the monthly mean and also allow for the construction of a co-located time series with the Level 2 satellite data. This could help highlight some of the causes of model biases by being able to pinpoint time and space where they occur more precisely. Model
230 biases may also result from the CMIP experimental design, such as requiring the climate state to be in equilibrium in 1850 while the real world may not have been (Bronselaer et al., 2017), or the parameterizations of biological and physical processes not allowing the system to change rapidly enough (Hoffman et al., 2014). Along with a longer time series, newer satellites, such as OCO-2 or the planned Sentinel 7 bring higher resolutions and more data, potentially helping to fill in the gaps and reduce the impact of the sampling we discussed in Section 5.3.2.".

235

**Specific comments**
**• Abstract: The abstract is quite long and technical. Describe the major findings concisely and leave nonessential details to the main text.**
240 We have shortened the abstract by removing several of the technical sentences which are not required to understand the results.

**• P1L20: The multi-model means give a feeling that CMIP6 models had a great improvement relative to CMIP5 models. But a closer look at Fig. 3 would tell that this was mainly because negative biases canceled positive biases. I**
245 **suggest adding the multi-model standard deviations of the bias (or other statistics that characterize the spread) in parentheses.**
We have added the full model spread alongside the means.

**• P1L25–32: The "trend reversal" in SCA caused by sampling bias is not clear on a first reading. It would be helpful**
250 **to write this in a way that is less entangled.**
We have rewritten this part as follows to make it clearer: "While the combined satellite product shows a strong negative trend of decreasing effective SCA with increasing $XCO_2$ in the northern midlatitudes, both CMIP ensembles instead show a non-significant positive trend in the multi-model mean. The negative trend is reproduced by the models when sampling them as the observations, attributing it to sampling characteristics. Applying a mask of the mean data coverage of each satellite to
255 the models, the effective SCA is higher for the SCIAMACHY/ENVISAT mask than when using the TANSO-FTS/GOSAT mask. This induces an artificial negative trend when using observational sampling over the full period, as SCIAMACHY/ENVISAT covers the early period until 2012, with TANSO-FTS/GOSAT measurements starting in 2009."

• **P2L46–60: If the purpose of this paragraph is to introduce SCA, then the first few sentences seem quite redundant. Better get to the point straight away.**

We have rewritten this part to be a more concise introduction of SCA: "Photosynthesis causes a net uptake of atmospheric $CO_2$ and thus declining atmospheric $CO_2$ concentrations in the growing season. Conversely, atmospheric $CO_2$ concentrations rise throughout the dormant season when there is a net release of $CO_2$ from the land due to decomposition of organic matter in soils. This uptake and release of carbon by the terrestrial biosphere throughout the year causes a seasonal cycle of atmospheric $CO_2$ (Keeling et al., 1989)."

• **P2L64: Missing Ciais et al., 2013 in the References.**

Fixed issue which resulted in reference not showing in the bibliography, thank you for spotting this.

• **P3L65: "downlooking" → "downward-looking"**

Changed.

• **P3L89–90: "such as a general overestimation of photosynthesis [relative to datadriven models]." We don't know the magnitude of the global photosynthesis with certainty. The number could range from 112 to 169 PgC yr-1 (Ryu et al., 2019, RSE). The MTE GPP data product (Jung et al., 2011, JGR) that was used to evaluate the CMIP models would sit near the lower end of this range.**

We have added this as a caveat and added the example of the overestimated leaf area index." such as a general overestimation leaf area index and photosynthesis. However, the magnitude of the global photosynthesis is not well constrained by observations, with estimates ranging between 112 and 169 PgC yr$^{-1}$ (Ryu et al., 2019), with the dataset used by Anav et al. (2013) is on the lower end of this range."

**P5L131: Missing Dlugokencky et al., 2018 in the References.**

Updated this reference to newer version and solved issue preventing it from showing in bibliography, thank you for spotting this.

• **P7L203: "parameters" → "variables"**

Changed.

• **P7L205: Surface observations sample the air within or closer to the boundary layer, and therefore may have a larger seasonal swing.**

The larger seasonal swing and thus higher seasonal cycle amplitude at the surface is discussed in the next paragraph. L205 was referring to the total offset of the mean, as evident most clearly in the Cape Grim and Hegyhatsal stations, comparing the blue (station) and red (model surface) curves.

• **P8L241: What is the correlation between the observed GR and the multi-model mean GR?**

The correlation between the observed and multi-model mean annual GR is 0.48 in CMIP6 and 0.07 in CMIP5, which is higher than most the models. We have added this to the paragraph.

• **P9L256: The increase of GR sensitivity to temperature after including 2015 and 2016 data could have been due to El Niño.**

This is true and has been added as a note in this sentence (now in the Appendix), as well as mentioned as a reason for suggesting that the timeseries is not long enough to support trying to reproduce this emergent constraint.

• **Table 1: Use whole numbers in the "Altitude" column.**

Changed to whole numbers.

• **Figure 2: I suggest lightening the gray background to prevent it from interfering with the reading of the curves.**

Changed the color of the land areas of the underlaying map plot to a very light grey.

310

**• Figure 3: It seems that after detrending, the seasonal variability in the multimodel mean XCO2 would match quite well with that in the observed XCO2, and better than the seasonal variability in any individual models. I wonder what the correlation coefficients would be.**
We have extended this figure with panels showing the calculated monthly growth rate and the detrended seasonal cycle. The
315 correlation between the detrended seasonal cycle of the multi-model mean to the observations is 0.98 in CMIP6 and 0.93 in CMIP5. in CMIP6 this is the best correlation, the closest models have a correlation of 0.96 (MRI-ESM2-0 and GFDL-ESM4). For CMIP6 a few models are close and a bit higher than the multi-model mean with 0.93 for NorESM1-ME and MIROC-ESM, 0.94 for MRI-ESM1 and 0.95 for MPI-ESM-LR. So the multi-model mean in CMIP6 indeed captures the seasonal variability better than any individual model.

320

**• Figure 4: I think ranking the models by their average growth rates, from low to high, would make the figure clearer.**
Models are now ranked from low to high in the barplot.

325 **• Figures 6a and 6b: Why not use the same scale? One could reserve the purple region of the colormap for the high SCA values from MPI-ESM-LR. This would not affect other models that have values represented in colors from blue to red.**
We have adjusted the scale to be the same for both figures. We have furthermore changed the colormap to a non-diverging one, which was requested by Referee #1 for the top panels in figure 6 (formerly 7) to contrast the bottom panels showing the
330 differences. This colormap has been adapted for Figure 5 (formerly 6) as well for consistency.

[4]LMD/IPSL, ENS, PSL Université, École Polytechnique, Institut Polytechnique de Paris, Sorbonne Université, CNRS, Paris, France

*Correspondence to*: Bettina K. Gier (gier@uni-bremen.de)

**Abstract.** Earth System Models (ESMs) participating in the Coupled Model Intercomparison Project Phase 5 (CMIP5) showed large uncertainties in simulating atmospheric $CO_2$ concentrations.  We utilize the Earth System Model Evaluation Tool (ESMValTool) to evaluate emission driven CMIP5 and CMIP6 simulations with satellite data of column-average $CO_2$ mole fractions ($XCO_2$). XCO₂ time series show a large spread among the model ensembles both in CMIP5 and CMIP6. Compared to the satellite observations, the models have a bias of +25 to -20 ppmv in CMIP5 and +20 ppmv to -15 ppmv in CMIP6, with the multi-model mean biases at +10 ppmv and +2 ppmv respectively. The derived mean atmospheric $XCO_2$ growth rate (GR) of 2.0 ppmv yr$^{-1}$ is overestimated by 0.4 ppmv yr$^{-1}$ in CMIP5 and 0.3 ppmv yr$^{-1}$ in CMIP6 for the multi-model mean, with a good reproduction of the interannual variability. All models capture the expected increase of the seasonal cycle amplitude (SCA) with increasing latitude, but most models underestimate the SCA. Any SCA derived from data with missing values can only be considered an "effective" SCA, as the missing values could occur at the peaks or troughs. The satellite data are a combined data product covering the period 2003–2014 based on the SCIAMACHY/ENVISAT (2003–2012) and TANSO-FTS/GOSAT (2009–2014) instruments.  While the combined satellite product shows a strong negative trend of  decreasing effective SCA with increasing XCO₂  in the northern midlatitudes, both CMIP ensembles  instead show a non-significant positive trend in the  multi-model mean  The  Most models from both ensembles show a positive trend of the SCA over the period 2003–2014, i.e. an increase of the SCA with XCO₂, similar to in

situ ground-based measurements. In contrast, the combined satellite product shows a negative trend over this period. Any SCA derived from sampled data 
[revised manuscript text omitted]
$^{-1}$ and the spread has decraseddecreased by 0.6 ppmv yr$^{-1}$, with a range from $1.7 \pm 0.4$ ppmv yr$^{-1}$

660    (MRI-ESM2.0) to $2.6 \pm 0.7$ ppmv yr$^{-1}$ (GFDL-ESM4). As expected from Figure 3, the models - with the exception of MRI-ESM1and, MRI-ESM2.0 and MIROC-ES2L - overestimate the growth rate, leading to an increased XCO$_2$ level in the present-day atmosphere compared to observations. The interannual variability of the growth rate for the models is generally higher than that of the observations, but well reproduced in the multi-model mean.

Emergent Constraints are relationships defined using an ensemble of models, between a measurable aspect of current or past

665    climate and an aspect of Earth system feedback in the future, which can be constrained using observational data (Eyring et al., 2019b). In Appendix C we have tried to reproduce the trend in interannual variability (IAV) of CO$_2$ growth rate to IAV of tropical temperature used by Cox et al. (2013) to develop an emergent constraint on the sensitivity of tropical land carbon to climate change, but were unable to find a significant trend for this much shorter satellite-derived time series.

The spatial variability of the GR is small, as $CO_2$ is long-lived and well mixed in the atmosphere with a one year mean interhemispheric crossing time. Thus there should be no significant regional changes on an annual level. Buchwitz et al. (2018) found the growth rate of the satellite dataset to be in agreement with those quoted by NOAA ESRLs global and Mauna Loa time series, as well as robust over several latitude bands. Our own analysis also shows only very small regional differences in the growth rate (not shown). No significant changes to the annual growth rates due to the satellite spatial coverage were found. ~~Emergent Constraints are relationships defined using an ensemble of models, between a measurable aspect of current or past climate and an aspect of Earth system feedback in the future, which can be constrained using observational data (Eyring et al., 2019b). Cox et al. (2013) developed an emergent constraint on the sensitivity of tropical land carbon to climate change using the sensitivity of the interannual variability (IAV) of $CO_2$ growth rate to the IAV of tropical temperature, which was later adapted by Wenzel et al. (2014) to CMIP5 models. Figure 5 shows the sensitivity of the IAV of $XCO_2$ growth rate to the tropical growing season temperature IAV for CMIP6 (left) and CMIP5 (right) models, both compared with observations. The observational temperature is taken from the GISTEMP v4 dataset (Lenssen et al., 2019) and the models use their own modeled temperature. We find an observational value of -0.23 ± 0.70 ppmv yr⁻¹ K⁻¹ for the 2003-2014 period. However, when using the full span of the satellite data until 2016, the slope increases to 0.75 ± 0.6 ppmv yr⁻¹ K⁻¹ (not shown), as the additional years show both a high growing season temperature and GR IAV. This shows that the time period 2003-2014 is not sufficient to reproduce the emergent constraint, but it might be feasible once CMIP6 emission driven future simulations are available for a longer time overlap between models and observations. CMIP5 model values for the timeframe 2003-2014 range from 0.53 ± 0.51 ppmv yr⁻¹ K⁻¹ (NorESM1-ME) to 3.14 ± 0.63 ppmv yr⁻¹ K⁻¹ (MRI-ESM1), with only CESM1-BGC showing a negative trend of -0.64 ± 0.55 ppmv yr⁻¹ K⁻¹. The multi-model mean has a value of 0.23 ± 0.70 ppmv yr⁻¹ K⁻¹. In CMIP6 the range is significantly decreased with a minimum of 1.14 ± 0.56 ppmv yr⁻¹ K⁻¹ (ACCESS-ESM1.5) to a maximum of 2.07 ± 0.33 ppmv yr⁻¹ K⁻¹ (CanESM5), with the multi-model mean at 1.36 ± 0.32 ppmv yr⁻¹ K⁻¹.~~

[revised manuscript text omitted]

**Figures**

[Figure]

Figure 1: Mean fractional coverage of monthly satellite data for 2003–2014. A value of 0 (white) signifies no available data, while a value of 1 (dark green) means that this gridcell contains data for all years of this month.

[Figure]

[Figure]

Figure 2: Comparison of time series from satellite $XCO_2$ (black), multi-model mean $XCO_2$ (orange) and surface $CO_2$ (red), and NOAA surface $CO_2$ station data (blue) at selected sites, with the coordinates noted in brackets above the time series and the altitudes shown in the map plot (see table 1). The multi-model mean for both $XCO_2$ and surface $CO_2$ was offset to have the same average value as the satellite $XCO_2$ for better comparison, and this offset is noted above each time series. CMIP6 and CMIP5 multi-model mean are shown on the top (a) and bottom (b) panels respectively.

[Figure]

[Figure]

**Figure a**: Global time series of monthly mean column averaged carbon dioxide (XCO₂) from 2003 to 2014, for the emission driven CMIP6  model simulations in comparison to satellite XCO₂ data (bold black line). The model output is sampled as the satellite data. The top panels show the time series, while the middle panels show the computed monthly growth rate, which has been used to detrend the data to obtain the seasonal cycle shown in the bottom panel. All available ensemble members for each model are shown to show the intrinsic variability of the models.

[Figure]

[Figure]

 **Figure 3b: Same as Figure 3a but for CMIP5. Only one ensemble member is shown.**

[Figure]

**Figure 4.** Global mean and standard deviation over all years of annual growth rate of XCO₂ during 2003–2014, for CMIP6 models (a) and CMIP5 models (b). The black bar  represents the satellite observations, while the red bar  depicts the multi-model mean.

1350

[Figure]

**Figure 5: Sensitivity of the interannual variability of the XCO₂ growth rate in the tropics (30° S-30° N) to the interannual variability of tropical growing season temperature for CMIP6 models (a) and CMIP5 models (b). The observational temperature taken from the GIStemp temperature anomaly map, while the models use their own simulated temperature. A linear regression is performed on the data for each dataset. Model colors are the same as in Figure 3, and symbols denote the years. In the top left of each panel the regression coefficient and its uncertainty is shown, while the bottom right states the correlation coefficient.**

**(a) CMIP6**

[Figure]

[Figure]

Figure 6a5a: Maps of mean annual seasonal cycle amplitude for 2003–2014 for CMIP6 models. The model name is given at the top left of each panel, while the top right shows the global average of the mean annual seasonal cycle. The panel to the right of the maps shows the zonal mean SCA.

[Figure]

[Figure]

1365  **Figure** **5b**: **Same as Figure** **5a** **but for CMIP5 models.**

[Figure]

[Figure]

**Figure 7a6a: Maps of mean SCA of the CMIP6 multi-model mean for 2003–2014. Top: SCA of multi-model mean with observational sampling (left) and without sampling (right). Middle: SCA of the satellite observations. Bottom: Difference between observations and sampled model data (left) and sampled and unsampled model (right).**

(b) CMIP5

[Figure]

[Figure]

**Figure 7b6b: Same as Figure 7a6a but for the CMIP5 multi-model mean.**

[Figure]

[Figure]

Figure 87: Trend of SCA with XCO₂ 2003–2014 for the northern mid-latitudes (30–60° N), including a linear regression with slope and mean SCA given on the top left of each panel and the Pearson correlation coefficient as well as the p-value on the bottom right. Symbols denote the different years and model colors are consistent with previous figures. The left panels (a) show unsampled CMIP6 models, while CMIP6 models sampled according to the satellite data are shown on the right (b). Note that the y-axis range for each plot is the same and only differs by a shift.

[Figure]

**Figure 98: Data Coverage of the satellite observations for (a) 2003–2008, containing only SCIAMACHY data, (b) 2009–2012 containing the overlap of SCIAMACHY and GOSAT data, and (c) 2013–2014 containing only GOSAT data. The patterned area highlights values above 0.5.**

1390

[Figure]

[Figure]

**Figure 910: Same as Figure 87, but with CMIP6 models masked using (a) the SCIAMACHY mask and (b) the GOSAT mask, with the masks derived from Figure 98, masking out points with less than 50% coverage in those time periods.**

1395

[Figure]

[Figure]

**Figure B1: Same as Figure 8̶7 but for CMIP5 models. The left panels (a) show unsampled models, while models sampled according to the satellite data are shown on the right (b). Note that the y-axis range is the same and only differs by a shift.**

1400

[Figure]

[Figure]

**Figure B2: Same as Figure B1, but with CMIP5 models masked using (a) the SCIAMACHY mask and (b) the GOSAT mask, with the masks derived from Figure 98, masking out points with less than 50% coverage in those time periods.**

1405

[Figure]

**Figure C1: Sensitivity of the interannual variability of the $XCO_2$ growth rate in the tropics (30° S–30° N) to the interannual variability of tropical growing season temperature for CMIP6 models (a) and CMIP5 models (b). The observational temperature taken from the GIStemp temperature anomaly map, while the models use their own simulated temperature. A linear regression is performed on the data for each dataset. Model colors are the same as in Figure 3, and symbols denote the years. In the top left of each panel the regression coefficient and its uncertainty is shown, while the bottom right states the Pearson correlation coefficient and p-value.**